# Stem cell-associated heterogeneity in Glioblastoma results from intrinsic tumor plasticity shaped by the microenvironment

Anne Dirkse[1,2,12], Anna Golebiewska [1,12], Thomas Buder[3,4], Petr V. Nazarov [5], Arnaud Muller[5], Suresh Poovathingal[6], Nicolaas H.C. Brons[7], Sonia Leite[8], Nicolas Sauvageot[8], Dzjemma Sarkisjan[1], Mathieu Seyfrid[1], Sabrina Fritah[1], Daniel Stieber[1], Alessandro Michelucci [1,6], Frank Hertel[9], Christel Herold-Mende[10], Francisco Azuaje [5], Alexander Skupin[6], Rolf Bjerkvig[1,11], Andreas Deutsch[3], Anja Voss-Böhme[3,4] & Simone P. Niclou [1]

The identity and unique capacity of cancer stem cells (CSC) to drive tumor growth and resistance have been challenged in brain tumors. Here we report that cells expressing CSC-associated cell membrane markers in Glioblastoma (GBM) do not represent a clonal entity defined by distinct functional properties and transcriptomic profiles, but rather a plastic state that most cancer cells can adopt. We show that phenotypic heterogeneity arises from non-hierarchical, reversible state transitions, instructed by the microenvironment and is predictable by mathematical modeling. Although functional stem cell properties were similar in vitro, accelerated reconstitution of heterogeneity provides a growth advantage in vivo, suggesting that tumorigenic potential is linked to intrinsic plasticity rather than CSC multipotency. The capacity of any given cancer cell to reconstitute tumor heterogeneity cautions against therapies targeting CSC-associated membrane epitopes. Instead inherent cancer cell plasticity emerges as a novel relevant target for treatment.

[1] NorLux Neuro-Oncology Laboratory, Department of Oncology, Luxembourg Institute of Health, L-1526 Luxembourg, Luxembourg. [2] Faculty of Science, Technology and Communication, University of Luxembourg, L-4365 Esch-sur-Alzette, Luxembourg. [3] Zentrum für Informationsdienste und Hochleistungsrechnen, Technische Universität Dresden, D-01069 Dresden, Germany. [4] Fakultät Informatik/Mathematik, Hochschule für Technik und Wirtschaft Dresden, D-01069 Dresden, Germany. [5] Proteome and Genome Research Unit, Department of Oncology, Luxembourg Institute of Health, L-1526 Luxembourg, Luxembourg. [6] Luxembourg Centre for Systems Biomedicine, University of Luxembourg, L-4367 Belvaux, Luxembourg. [7] National Cytometry Platform, Luxembourg Institute of Health, L-1526 Luxembourg, Luxembourg. [8] Centre of Competence for Methodology and Statistics, Luxembourg Institute of Health, L-1445 Strassen, Luxembourg. [9] Department of Neurosurgery, Centre Hospitalier Luxembourg, L-1210 Luxembourg, Luxembourg. [10] Division of Neurosurgical Research, Department of Neurosurgery, University of Heidelberg, 69120 Heidelberg, Germany. [11] Department of Biomedicine, University of Bergen, N-5019 Bergen, Norway. [12] These authors contributed equally: Anne Dirkse, Anna Golebiewska. Correspondence and requests for materials should be addressed to S.P.N. (email: simone.niclou@lih.lu)

Glioblastoma (GBM) displays extensive cellular heterogeneity which represents a major obstacle for effective treatment. Similar to other cancers, tumor progression has been proposed to rely on cancer stem cells (CSC), responsible for tumor recurrence and resistance to therapy. CSCs are postulated to display diverse stem cell properties and to be highly tumorigenic in experimental models in vivo[1]. The model predicts that CSCs reside at the apex of a hierarchical organization and recreate intra-tumoral phenotypic heterogeneity by generating differentiated progeny. Recent single-cell transcriptomic analysis revealed stem cell-signatures to be associated with the most proliferative cells in low grade gliomas, where stemness increases with tumor grade[2,3]. Such an organization was less clear in GBM, which displayed a continuum of stemness profiles anti-correlated with cell-cycle genes[4]. Although very informative, such data describe marker expression at a given snapshot in time and do not consider the dynamic functional properties of tumor cells displaying different phenotypes. Similarly, genetic barcoding techniques suggesting a proliferative hierarchy in GBM[5] cannot address phenotypic heterogeneity and evolution of phenotypic states over time.

Identification of CSCs is largely based on the expression of cell membrane antigens, which are amenable to targeted therapy[6]. In GBM many studies rely on cell surface markers such as CD133, CD15/SSEA, CD44, or A2B5 for CSC isolation[7–10], yet no single marker is able to define a universal GBM CSC population[11]. The identity of GBM CSCs is still unresolved and, although widely used, there is controversy whether marker-expressing cells fulfill the functional criteria of bona fide CSCs[12] and whether CSCs represent a quiescent or a proliferative subpopulation. In this context, functional assays combined with marker expression are indispensable for the validation of CSC properties[1].

The hierarchical CSC model has been challenged by growing evidence suggesting that CSCs may not constitute a defined cellular entity, but rather a cellular state adapting to microenvironmental cues[13]. Initial reports on GBM suggested that only CSC-marker positive cells were able to form tumors[7,9], while later studies reported either no difference in tumorigenic potential[8,14,15] or both fractions being tumorigenic, but with different potency[11,16,17]. Although generally marker positive cells were shown to be multipotent, multipotency of marker negative cells was rarely addressed. Several GBM studies, however, showed that marker positive cells can be derived from the negative fraction and regain the initial heterogeneity[11,14,17,18] supporting strong tumor plasticity in recreating intra-tumoral phenotypic heterogeneity. Numerous data supporting the concept of plasticity[19,20] point to a role of the microenvironment in shaping the phenotype toward spatial and temporal heterogeneity[21]. Indeed, GBM cells expressing stem cell markers are often attributed to specific tumor niches[22–26]. It still remains unclear whether the microenvironment selects for survival of specific CSCs or whether tumor cells adapt within new microenvironments. Intriguingly, recent data further showed that GBM CSCs alone carry limited tumorigenic potential, and reciprocal crosstalk with tumor cells representing more differentiated phenotypes creates supportive niches and promotes tumor growth[27,28]. These results point toward a key role of tumor cell plasticity and intra-tumoral phenotypic heterogeneity in shaping tumor progression. Although several mathematical approaches aimed to model the creation and maintenance of CSC and non-CSC states in developing tumors based on intrinsic cellular parameters, modeling tumor plasticity in the context of changing spatial and temporal microenvironments remains challenging[21,29,30].

Here we asked whether cancer cells expressing CSC-associated cell membrane markers are a defined entity at the apex of a hierarchical organization or whether they represent a phenotypic state that cells can reversibly acquire in response to environmental cues. Similar to patient biopsies, we observe CSC markers to be heterogeneously expressed in patient-derived GBM xenografts and stem-like cell cultures. We find that all GBM cell subpopulations that carry stem cell properties, are tumorigenic and are able to adapt to various environmental changes. Our data show that cells expressing stem cell markers do not represent a clonal entity defined by distinct functional properties and transcriptomic signatures, but rather a cellular state that is determined by environmental conditions. Such states are non-hierarchical, reversible and occur via stochastic state transitions of existing populations, striving toward an equilibrium within a given microenvironment. While all subpopulations survive and are capable of phenotype adjustment, differences exist in the adaptation speed resulting in different tumor growth rates in vivo. This indicates that the outcome and interpretation of functional assays depend on dynamic processes and endpoint of analysis. Our data provide evidence that stem cell-associated phenotypic heterogeneity in GBM is a result of intrinsic cancer cell plasticity, which has important implications for the design of treatment strategies. If the CSC state is an inducible and transient state, then targeting only a small subpopulation of cancer cells will be ineffective, instead the dynamic processes will need to be tackled.

## Results

**Inter- and intra-tumor heterogeneity of putative CSC marker.** So far no universal CSC marker has been identified in GBM. In an attempt to identify cell surface markers shared by most GBM patients, we compared the expression of 10 commonly used CSC-associated markers in the TCGA patient cohort. This showed variable expression levels across tumors (Fig. 1a) and little association with known genetic or epigenetic alterations (Supplementary Fig. 1A–C). No significant correlations were detected between marker genes (Fig. 1b). As expected[31], several markers were enriched in previously defined transcriptional subgroups (Supplementary Fig. 1D, E).

It is well established that gene expression does not necessarily correspond to epitope presentation and CSCs may represent varying abundance across individual patient tumors[32]. Moreover, bulk analysis of transcriptional profiles in GBM patients represents mean values of tumor and stroma[33] and cell membrane epitopes can be present in both compartments[34]. Therefore, we applied multicolor flow cytometry on fresh GBM tissue isolated from 11 patient-derived orthotopic xenografts (PDOXs). PDOXs were established from GBM with different genetic background[35] in eGFP-expressing mice[36], which allowed us to discriminate tumor and stromal cells and phenotype the tumor compartment in an unbiased manner (Supplementary Fig. 2A, B). Similar to gene expression analysis, epitope presentation of 12 CSC markers was highly variable between tumors (Fig. 1c, Supplementary Fig. 3A, B). CD90 and CD29 were strongly positive in most GBM, whereas CD24 and CD195 were largely absent. In most cases, marker expression was relatively homogeneous within one GBM, while only 4/12 markers (CD133, CD44, CD15, and A2B5) showed relevant intra-tumoral heterogeneity (Fig. 1c, Supplementary Fig. 3B), which would be expected from a bona fide CSC marker. We have previously described different microenvironmental landscapes in GBM PDOX models, either displaying a purely infiltrative phenotype or showing extensive necrosis, hypoxia, and vascular leakage[35]. However, no link between marker expression profiles and histopathological tumor phenotypes was observed here (Supplementary Fig. 3).

**Genetically distinct clones adapt marker expression in vivo.** We have previously shown that intra-tumor heterogeneity in GBM is

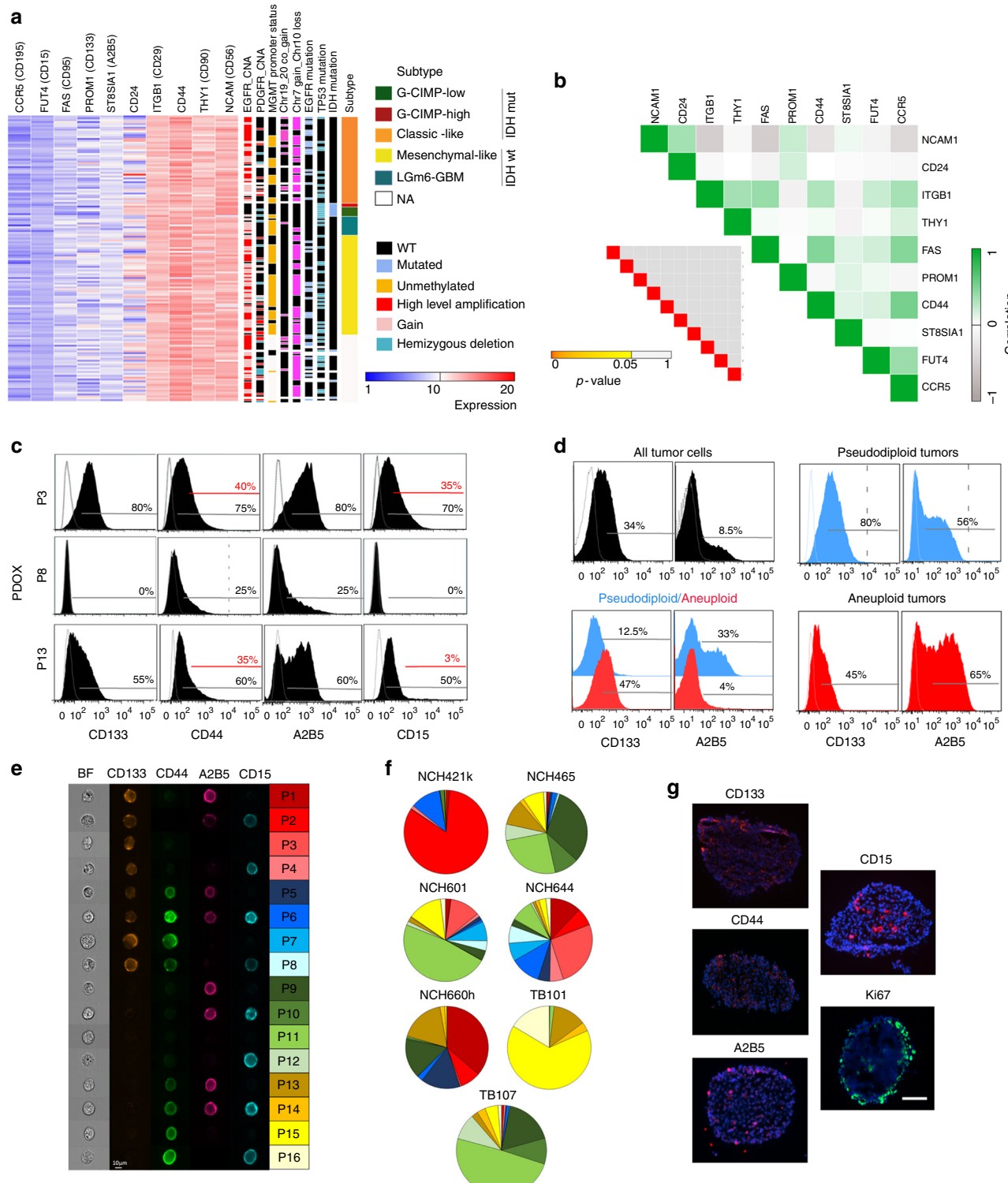

**Fig. 1** CSC-associated heterogeneity in GBM. **a** Inter-patient heterogeneity at the gene expression level for a panel of CSC-associated markers. See also Supplementary Fig. 1. **b** Pearson analysis did not reveal any significant correlation. **c** Flow cytometric analysis of tumor cells in GBM PDOXs. Percentage of positive cells (black gate) is indicated vs. negative control and vs. high expressing cells (red gate). See Supplementary Fig. 2 for gating strategy and Supplementary Fig. 3 for more examples. **d** Marker expression profiles in the genetically heterogeneous PDOX T16 (all tumor (black), pseudodiploid (blue), and aneuploid (red) clones). Separately implanted pseudodiploid (blue) and aneuploid clones (red) adapted the CSC-associated profiles in the xenograft (right). **e** Multicolor phenotyping. Representative ImageStream data shown for NCH644. Right panel: color code for 16 subpopulations applied in all figures. **f** Distribution of subpopulations in seven patient-derived GBM cultures (mean %, error bars ommitted for visualization purpose). **g** Distribution of CSC-associated markers and Ki67 proliferating cells in 3D spheres (scale bar = 100 μm)

also present at the ploidy level and that CSC-associated marker expression is distributed throughout genetically different clones[37], as shown here, e.g., for CD133 and A2B5 (Fig. 1d). FACS-purified pseudodiploid and aneuploid cells independently gave rise to tumors in the mouse brain, with the aneuploid clone being more tumorigenic[37]. Here we assessed whether the differential marker profile was retained in individual ploidy clones or whether clones adapted their marker expression upon in vivo growth. Interestingly, both clones changed marker expression in vivo, i.e., pseudodiploid clones retained a heterogeneous A2B5 profile and increased CD133, while aneuploid clones significantly increased A2B5 epitope presentation (Fig. 1d), indicating that CSC-associated marker expression in vivo is not a defined property of a genetic clone and that tumor cells change marker expression following clonal selection, most probably reflecting an adaptation to the new microenvironment.

**Heterogeneity in GBM stem-like cultures**. In order to correlate marker presentation with CSC functional properties, we turned to patient-derived GBM cultures. Although in vitro models carry inherent drawbacks, patient-derived cultures represent the best model to address the full spectrum of stemness properties and are commonly applied for CSC studies in human GBMs[38]. This is not possible in our PDOX models due to limited proliferation in vitro. Although short-term cultures derived from GBM xenografts can be useful[27], in our experience these cultures are genetically unstable (similar to long term cultures) and during first passages contain genetically divergent clones with different ploidy levels (Supplementary Fig. 2F). To avoid any bias in the outcome of functional studies[37] we focused our analysis on seven GBM stem-like cultures, which are genetically stable, show a similar genetic profile as patient tumors and retain stable CSC marker expression over passages. Similar to patient biopsies and PDOXs, GBM cultures displayed remarkable inter-patient heterogeneity of CSC markers (Supplementary Fig. 3B), which was stable across cell passaging. CD133, CD44, CD15, and A2B5 showed again the strongest heterogeneity within each GBM culture. Focusing on these four markers, we performed multicolor flow cytometry to discriminate 16 subpopulations, which were applied to define phenotypic heterogeneity throughout the study (labeled P1–P16; Fig. 1e, f, Supplementary Fig. 2C, D). Some cultures contained a limited number of predominant subpopulations, while 4/7 cultures contained a substantial amount of all 16 subpopulations. The heterogeneity was present at the single sphere level without a particular localization pattern, nor a link with proliferative cells more prominent at sphere edges (Fig. 1g). All subpopulations were proliferating, although some CD133⁻CD44⁻ cells (P9–P12) contained less cells in S/G2/M (Supplementary Fig. 4A). This may reflect the fluctuation of CD133 across the cell cycle[39]. Thus, patient-derived GBM cultures recapitulate the intra-tumoral phenotypic heterogeneity of CSC marker expression observed in patient biopsies and PDOXs.

**All GBM subpopulations carry stem cell properties in vitro**. To investigate stem cell properties of different phenotypic states we combined FACS-based purification with functional assays. Self-renewal, proliferation, and multipotency tests were performed on the 16 subpopulations of highly heterogeneous GBM cultures (Fig. 2a). Importantly, all FACS-sorted subpopulations were subjected to the same culture conditions, giving them equal chance to display stem cell properties. This is in contrast to other studies that apply different growth conditions on marker positive (serum-free) and negative (serum-containing) subpopulations (e.g., refs. [25,27]), thereby introducing a bias induced by environmental factors. We found that all FACS-sorted subpopulations

were able to self-renew over multiple passages with no significant differences between each other (Fig. 2b). No dilution of differentiated counterparts and progenitors was observed as indicated by similar sphere size. All subpopulations proliferated indefinitely at a similar rate (Fig. 2c). This indicated that phenotypically heterogeneous GBM cells have similar stem cell properties in vitro, suggesting a lack of hierarchical organization.

**GBM subpopulations undergo stochastic state transitions**. We next performed multipotency tests by multicolor phenotyping for FACS-purified subpopulations over time to reveal which subpopulations were responsible for creating the phenotypic heterogeneity. Surprisingly, none of the subpopulations maintained its original phenotype in time. Alluvial plots in Fig. 2d represent the dynamic evolution of phenotypic heterogeneity, where the proportions of phenotypic states are ordered from the most prominent states at the top to the least prominent states at the bottom at each time point (see Supplementary Fig. 5A for an alternative display of the data in column charts, where phenotypic states are displayed in the same order, P1–P16, at each time point and Supplementary Data 2A for statistics). All subpopulations were able to recreate the phenotypic heterogeneity consisting of different phenotypic states, indicating that none of them represented a unipotent differentiated phenotype as expected in a one-way hierarchical model. This was also true for clones derived from single cells (Supplementary Fig. 5B). Most phenotypic states were already reestablished after 20 days (D20), although the adaptation was dynamic over time and varied between subpopulations (Fig. 2d). However, at day 70 (D70), all 16 subpopulations were reconstituted in each sample suggesting a tendency toward the original phenotypic equilibrium. The multipotency of different subpopulations was confirmed in additional GBM cultures (Supplementary Fig. 4B).

To quantify state transitions in time between the different phenotypes and predict the time of equilibrium, we applied CellTrans, a stochastic compartment model based on the Markov chain[40], which represents an established tool for assessment of state transitions based on FACS data[41]. We assumed that the phenotypic state compositions could be determined by two main processes: (i) stochastic state transitions due to cellular plasticity and (ii) different proliferation rates reflecting selection. Since all subpopulations proliferated at a similar rate (Fig. 2c), this parameter could be excluded from the modeling. We acknowledge that cell–cell interactions are likely to play a role in state transitions. However, since marker heterogeneity was present at the single sphere level without a specific distribution pattern (Fig. 1g) and in the absence of known quantifiable data about inter-cellular interactions, we neglected spatial aspects to keep the model tractable and focused on key processes. The model estimated the occurrence of 175 of 240 possible direct state transitions (Fig. 2e, Supplementary Fig. 5C). The estimated transition matrix appeared irreducible with no bottlenecks, i.e., each phenotype could transit to other states either directly or through intermediate steps. No hierarchies or sub-hierarchies were detected between the phenotypes (Krackhardt hierarchy score = 0). Moreover, transitions between positive and negative states of one marker were independent of the other markers (Supplementary Table 1A). Interestingly, the theoretical equilibrium was very similar to the original composition (Fig. 2f), even though this parameter had not been taken into account for the modeling.

We next tested the validity of the model by comparing the predicted values with experimental data by designing two combinations of subpopulations (mix A and B, time = 0) predicted to reach the equilibrium in 39 days. The FACS-sorted

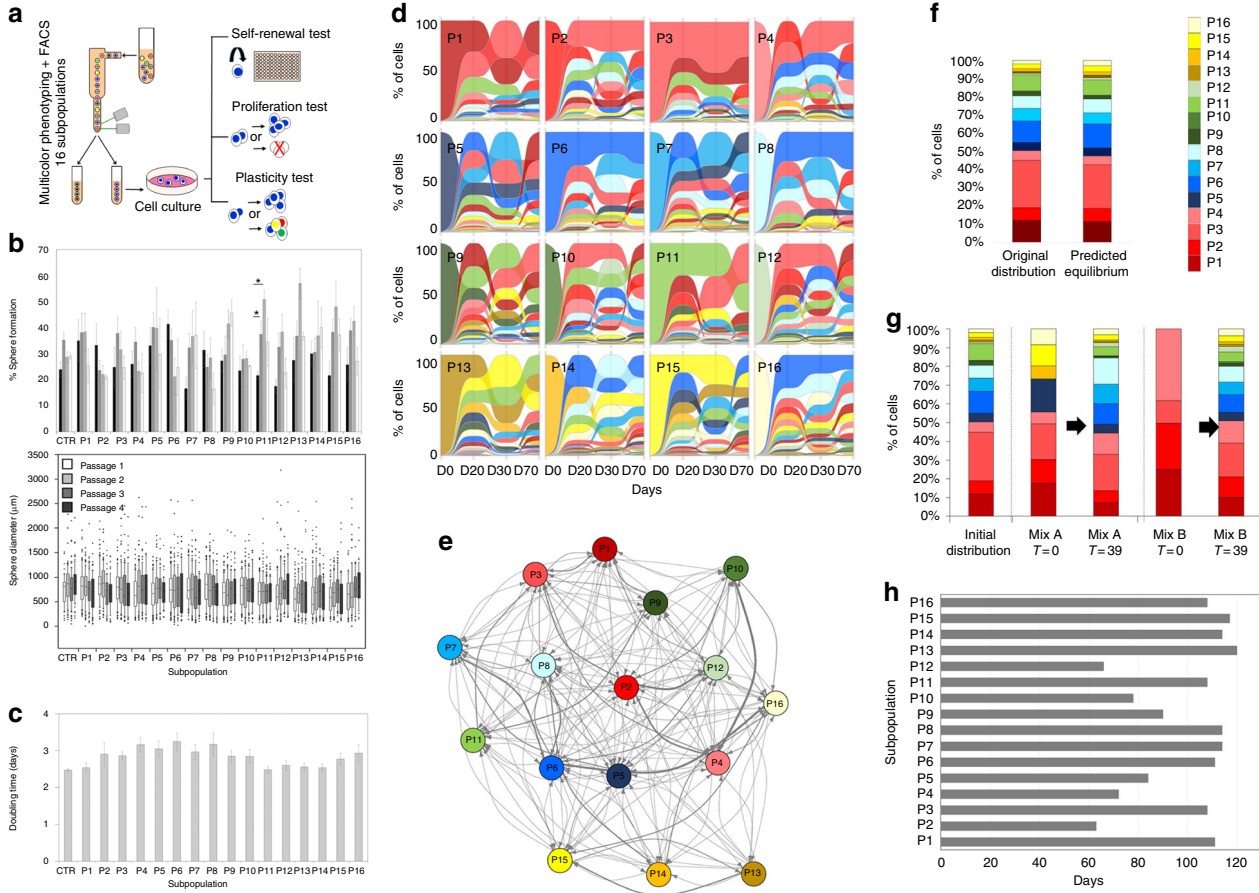

**Fig. 2** GBM subpopulations undergo state transitions in a non-hierarchical manner. **a** Experimental setup of FACS sorting and functional analysis performed on 16 subpopulations (NCH644). See Supplementary Fig. 2E for gating strategy. **b** Self-renewal test, including sphere formation (mean +/− SEM) and sphere diameter (Box limits indicate the 25th and 75th percentiles and center lines show the medians as determined by R software; whiskers represent the extreme low and high observed values, unless those are above 1.5 times interquartile range (IQR)—thereby whiskers are limited to 1.5 IQR. All outlying data points are represented by dots). Bulk cells were used as control (CTR). No statistical difference observed, except for P11 passages 1 vs. 2 and 1 vs. 3 (*$p$-value ≤ 0.05, Kruskal–Wallis test). **c** Proliferation test (mean doubling time +/− SEM). No statistical difference observed. **d** Multipotency test. Marker expression over time: FACS-sorting day 0 (D0) and re-phenotyping after 20 (D20), 30 (D30), and 70 (D70) days in culture. The order of subpopulations in alluvial plots is based on highest to lowest percentage at each time point. See an alternative representation in Supplementary Fig. 5A for column chart and Supplementary Data 2A for statistics (color code in Fig. 1e). **e** Markov modeling of state transitions between 16 subpopulations. Arrows represent predicted direct state transitions between subpopulations, thickness of lines corresponds to transition probabilities. See Supplementary Fig. 5C for transition matrix. **f** Proportions of subpopulations predicted in equilibrium state is similar to initial culture. **g** Validation of Markov modeling. FACS-sorted admixtures (time 0) were re-phenotyped at the predicted equilibrium time (39 days) showing the accuracy of the mathematical model. See Supplementary Data 1A for statistics. **h** Predicted time to reach equilibrium for each subpopulation

admixtures reverted to the original equilibrium at the predicted time point (Fig. 2g, Supplementary Data 1A). Interestingly, although the model predicted that all subpopulations reach the same equilibrium, they varied in the time needed (Fig. 2h), with P2 and P12 showing the fastest transitions. In conclusion, all subpopulations retain full capacity to generate other phenotypic states. Despite differences in pace, heterogeneity is recapitulated in time from each phenotypic state in a non-hierarchical manner.

**Hypoxia induces phenotypic adaptation.** We then asked to what extent the phenotypic heterogeneity is dependent on the microenvironment and whether changing conditions leads to selection or adaptation of phenotypic states. We first focused on hypoxia, a hallmark of GBM, representing a clinically relevant microenvironment. Although it has been reported that hypoxia enriches for CSCs[42,43], it is not clear whether such phenotypic shifts result from a selection of an existing CSC subpopulation or from phenotypic adaptation. Although GBM cells in vivo may be

subjected to an oxygen gradient depending on their spatial location, we opted to recapitulate severe hypoxia with oxygen levels < 0.5% $O_2$. This condition corresponds to the most hypoxic area surrounding the necrotic tumor zone, previously associated with the CSC niche. We have minimized the bias linked to acidification of culture media by regular monitoring of pH levels. Concordant with previous reports, exposure to hypoxia led to a clear shift in phenotype (Fig. 3a, Supplementary Fig. 4C, Supplementary Data 1B, C). For example, hypoxic NCH644 cells were more positive for CD133, CD44, and A2B5, whereas CD15 was decreased. In particular, subpopulations P5 and P7 were enriched, whereas several others (e.g., P2, P10) were reduced. Changes were not always gradual in time, suggesting the occurrence of indirect state transitions before reaching a new hypoxia-specific equilibrium. We have previously shown that low oxygen leads to growth inhibition, activation of glycolysis[44], and autophagy[45], although NCH644 displayed less dead/apoptotic cells compared with NCH421k (Supplementary Fig. 4F). The phenotypic changes were also reflected at the transcript level for

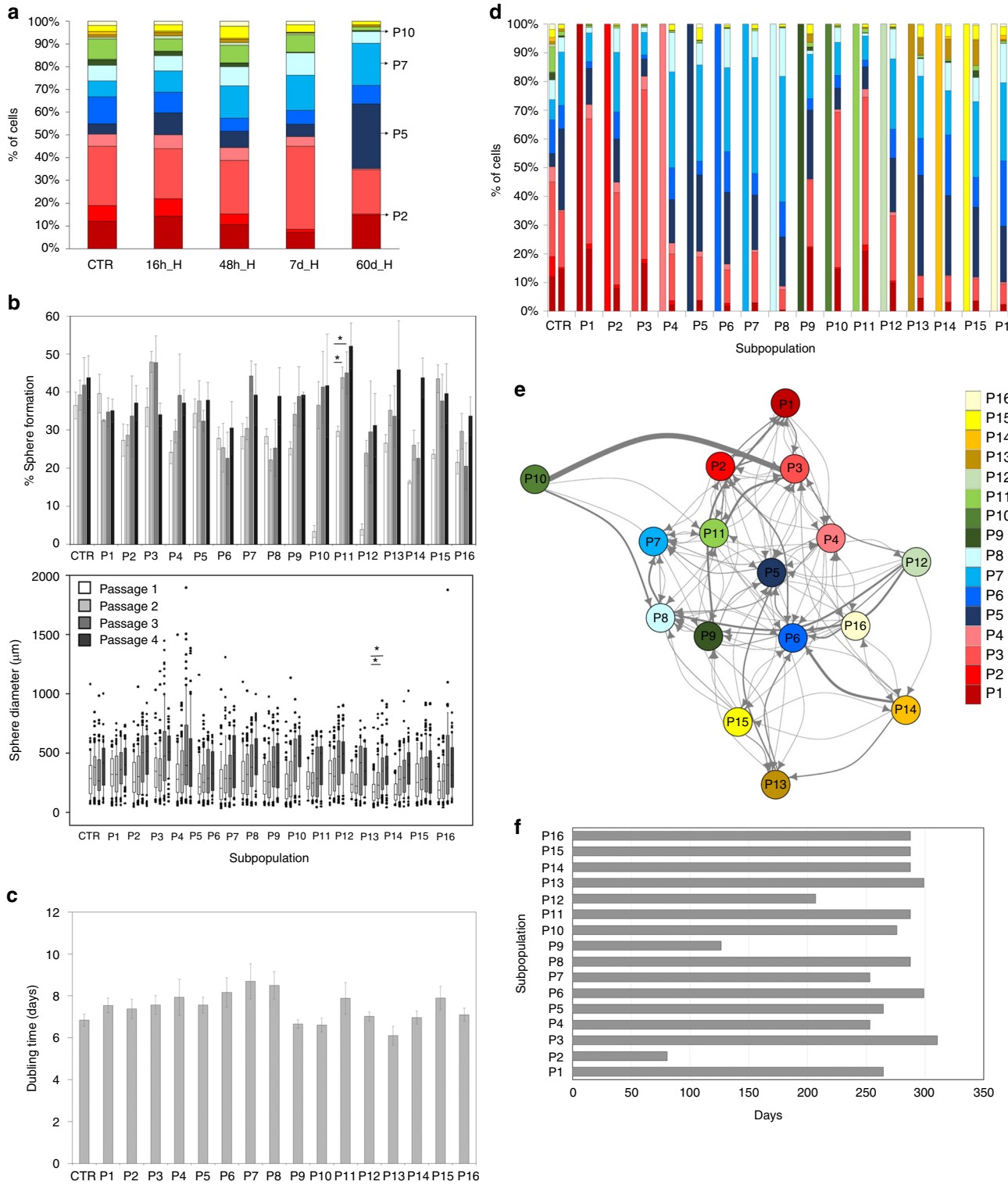

PROM1 and CD44 already after 12 h of hypoxia suggesting fast adaptation (Supplementary Table 2).

To test whether the changes were a result of selection or phenotypic adaptation we next performed functional assays under hypoxia (Fig. 2a). Again, all subpopulations self-renewed (Fig. 3b) and proliferated (Fig. 3c). Certain subpopulations differed in self-renewal potential during initial passages (Supplementary Data 4). P10, P12, and P14 showed low clonogenic potential at the first passage, which was in accordance with their

reduction in long-term hypoxia (Fig. 3a), pointing to a possible partial selection at early time points. These differences were later lost, indicating efficient adaptation. Although single cells gave rise to smaller spheres compared with normoxia, hypoxic spheres did not differ in size between each other (Fig. 3b). The decreased self-renewal did not correlate with the proliferation index as only two subpopulations statistically differed from each other (Fig. 3c). All subpopulations proliferated indefinitely, though at a decreased rate. The lack of correlation between phenotypes enriched in

**Fig. 3** Adaptation of GBM subpopulations to hypoxia. **a** Distribution of subpopulations in hypoxia (H). Normoxia (N) is shown as control (CTR) and hypoxia (H) at 16 h, 48 h, 7 days, and 60 days. Data shown for NCH644, additional cultures in Supplementary Fig. 4C. See statistics in Supplementary Data 1B, color code in Fig. 1e. **b** Self-renewal test of 16 subpopulations in hypoxia, including sphere formation (mean $+/-$ SEM) and sphere diameter (Box limits indicate the 25th and 75th percentiles and center lines show the medians as determined by R software; whiskers represent the extreme low and high observed values, unless those are above 1.5 times interquartile range (IQR)—thereby whiskers are limited to 1.5 IQR. All outlying data points are represented by dots). Bulk cells were used as control (CTR). Statistical differences within the same subpopulations are shown (*$p$-value $\leq 0.05$, Kruskal–Wallis test, see Supplementary Data 4B for statistics). **c** Proliferation test in hypoxia (mean doubling time $+/-$ SEM). Bulk cells were used as control (CTR). Statistical difference was only found between P7 and P13 (mixed linear model). **d** Distribution of 16 subpopulations at day 0 (left) and after 60 days in hypoxia (right) (see Supplementary Data 2B for statistics). **e** Markov modeling of state transitions in hypoxia. Arrows represent direct state transitions between subpopulations, thickness of lines corresponds to transition probabilities. See Supplementary Fig. 5E for transition matrix. **f** Predicted time needed to reach equilibrium for each subpopulation in hypoxia

hypoxia and their proliferation index suggests a strong role of phenotypic adaptation to a changing environment.

**Modeling GBM state transitions in hypoxia**. To model phenotypic adaptation to hypoxia, we analyzed FACS-sorted subpopulations after 60 days in hypoxia. All subpopulations created a phenotypic distribution resembling the original hypoxic cultures, rather than the normoxic equilibrium, e.g., CD133 levels were increased in all subpopulations regardless of their initial state (Fig. 3d, Supplementary Data 2B). Change toward a hypoxia-specific equilibrium of CD133 positive and negative cells was confirmed in other GBM cultures (Supplementary Fig. 4D). Statistical differences were still observed at 60 days, suggesting that the equilibrium had not been reached yet. Nevertheless, all subpopulations recreated a phenotypic heterogeneity in hypoxia. This was also true for clones reformed from single cells following the self-renewal test (Supplementary Fig. 5D, Supplementary Data 3B).

We have further applied Markov chain modeling to assess state transitions in hypoxia. Since all cells were subjected to severe hypoxia in vitro and subpopulations proliferated at similar rates, we applied the same modeling strategy as for normoxia, excluding the spatial aspect of microenvironmental cues. Markov chain modeling revealed that, although direct state transitions were more restricted compared with normoxia (103/240; Fig. 3e, Supplementary Fig. 5E), all subpopulations could transit to other states except to P10. This was in accordance with the very low proportion of P10 observed in hypoxia (Fig. 3a) and its low self-renewal potential at initial passage (Fig. 3b). Notably, P10 survived hypoxia and could transform to other states, preferentially to P3. Subpopulations enriched in hypoxia (e.g., P5, P7) could be formed from numerous phenotypic states ($\geq 9$). The transition matrix appeared reducible with one transient state (P10, i.e., no phenotype can transit into P10 and P10 is depleted from the hypoxic equilibrium), but no absorbing state (i.e., exit from this state is not possible). Most populations needed $> 250$ days to reach hypoxic equilibrium (Fig. 3f), compared with 92 days for the bulk culture. Of note, the subpopulations enriched in hypoxia (e.g., P5, P7) did not carry an advantage to reach equilibrium faster, rather, the subpopulations not enriched (P2, P9, and P12) were most adaptive. Interestingly, they also belonged to the fastest adapting subpopulations in normoxia (Fig. 2h). Altogether these results show that the phenotypic shift in hypoxia occurs through adaptation of existing cells. Only a very limited hierarchy (Krackhardt hierarchy score = 0.125, Supplementary Table 1B) was observed. Although we cannot exclude a partial selection for certain phenotypic states, all subpopulations survived hypoxia and re-adapted to reach a hypoxia-specific equilibrium.

**Induced phenotypic adaptations are reversible**. We further tested marker expression in more complex environments,

including combined exposure to extracellular matrix, differentiation agents (FBS, ATRA), and hypoxia. Upon differentiation cells underwent morphological changes both in normoxia (Fig. 4a) and hypoxia (Supplementary Fig. 6A). Although expression of neuronal (β-III-tubulin) and astrocytic (GFAP) markers increased, CSC-associated intracellular markers (Nestin, Vimentin) remained expressed, suggesting an incomplete differentiation process (Fig. 4b; Supplementary Fig. 6B). In normoxia, differentiation resulted in a strong shift toward CD133− and CD44+ subpopulations (P13–P16), although CD133−CD44− cells (P11) were also enriched (Fig. 4c, Supplementary Data 1D). Although similar changes were observed in hypoxia, the phenotypic shift was clearly the result of two environmental pressures, i.e., a decrease of CD133+ cells upon differentiation was compensated by increased CD133 in hypoxia (Fig. 4c). Similar data were obtained for another GBM model, which lost the predominance of the P2 subpopulation both in normoxia and hypoxia (Supplementary Fig. 6D, Supplementary Data 1D).

We next assessed the phenotypic heterogeneity upon returning to normoxic 3D conditions. Regardless of the differentiation status and oxygen level, GBM cells regrew as 3D spheres and regained the expression of intracellular stem cell markers (Supplementary Fig. 6C). This was accompanied by a partial regain of the initial cell membrane marker phenotype at day 14 (Fig. 4c, Supplementary Fig. 6D, Supplementary Data 1D, E). Markov modeling predicted a longer time to fully revert to normoxic equilibrium from differentiated states (75 and 84 days from Diff_N and Diff_H conditions, respectively). In summary, GBM cells undergo specific reversible phenotypic changes shaped concomitantly by various microenvironmental cues.

**Reversible adaptation of CSC-associated phenotypes in vivo**. We next asked whether the reversible phenotypic adaptation occurs in vivo. All five implanted GBM cultures adapted their phenotype to the brain environment (Fig. 4d; Supplementary Fig. 6E, F, Supplementary Data 1F, G) with a strong tendency toward the enrichment of CD133−CD44− cells (P9–P12) and depletion of CD133+CD44+ (P5–P8) and CD133−CD44+ (P13–P16) cells. These changes were also detected at the transcript level (Supplementary Table 2) and differed from the equilibria observed in hypoxic and differentiation conditions in vitro, suggesting the impact of additional factors in the complex in vivo microenvironment. This was also true for the intracellular stem cell and differentiation markers (Fig. 4e, Supplementary Fig. 6G). Again, the phenotypic change in vivo was reversible and xenografted tumor cells regained the heterogeneous profile when returned to in vitro conditions (Fig. 4d, Supplementary Fig. 6E, Supplementary Data 1F, G). Xenografted cells were recultured for a time period equivalent to the tumor development time, at which point the phenotypic equilibrium of the original cultures was not yet reached, in accordance with mathematical modeling.

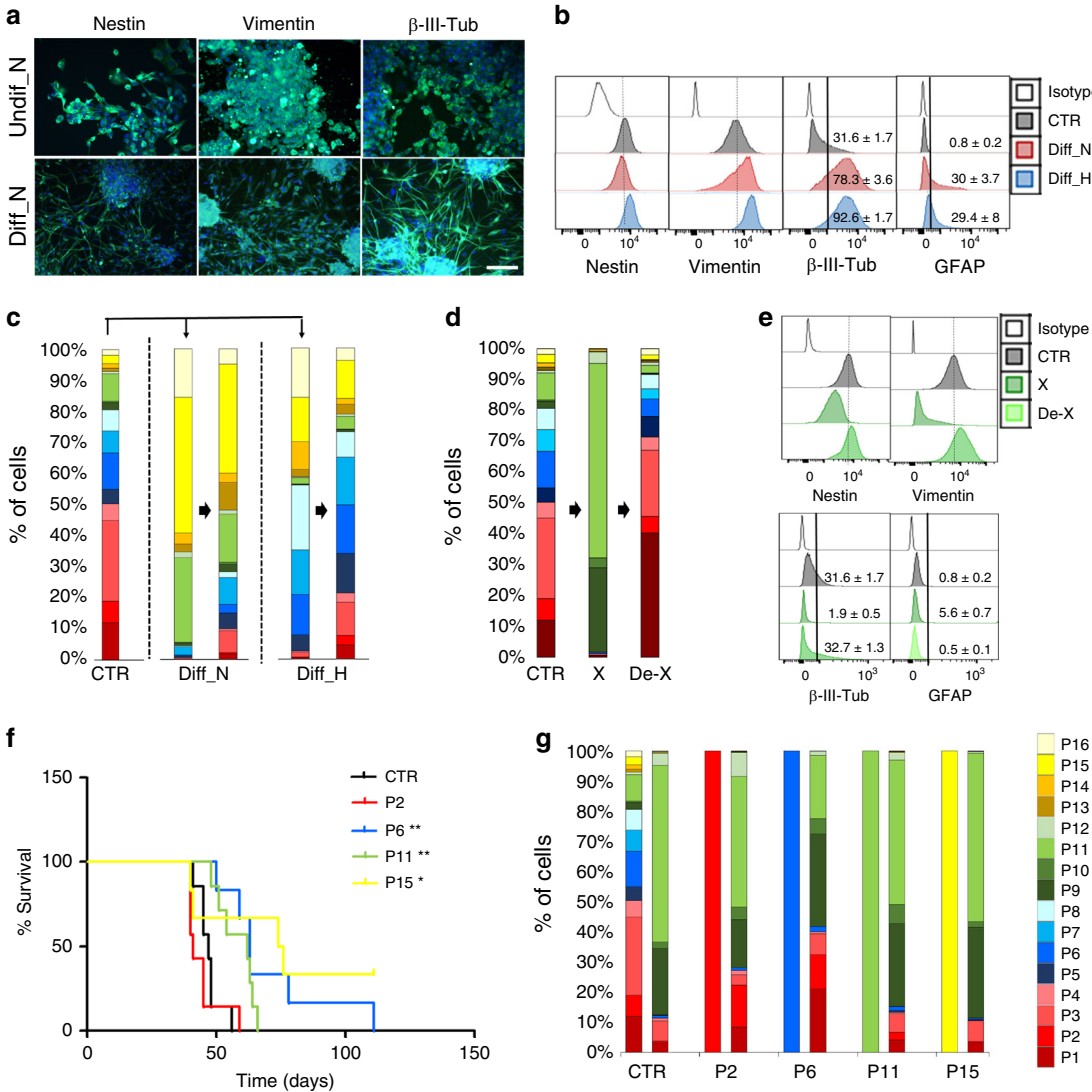

**Fig. 4** Reversible phenotypic changes. **a** Examples of adherent GBM cells in stem cell (Undif_N) or differentiation (Diff_N) conditions in normoxia. See Supplementary Fig. 6A for hypoxia (scale bar = 200 μm). **b** Flow cytometric analysis of intracellular markers in control 3D sphere cultures (CTR), differentiation (Diff) conditions. N normoxia, H hypoxia. Black lines discriminate between negative and positive cells (mean +/− SEM, n = 3). Dotted line indicates mode expression in control cells. See also Supplementary Fig. 6B, C. **c** Distribution of subpopulations under different environmental conditions, phenotyping performed after 14 days of change (left) and 14 days after reverting to control culture (right). See Supplementary Data 1D for statistics and Supplementary Fig. 6D for more examples. **d** Distribution of subpopulations in xenografted NCH644 tumors in vivo (X) and after regrowth in vitro (De-X). Normoxia cultures are shown as control (CTR). See Supplementary Data 1F for statistics and Supplementary Fig. 6E, F for more examples. **e** Flow cytometric analysis of intracellular markers. Black lines discriminate between negative and positive cells (mean +/− SEM, n = 3). Dotted line indicate mode expression in control cells. **f** Kaplan–Meier survival curves of xenotransplanted mice. Subpopulations P2, P6, P11, and P15 were implanted directly after FACS. FACS-sorted bulk cells were used as control (CTR) (*p-value ≤ 0.05; **p-value ≤ 0.01, long-rank test). **g** Distribution of subpopulations in xenografted tumors. For each subpopulation, day of implant and day of mouse sacrifice are presented. See Supplementary Data 1H for statistics

**Accelerated tumor growth in vivo of more plastic states.** To rule out a negative selection against certain subpopulations, we implanted individual subpopulations directly after FACS sorting, focusing on four phenotypic states: P6 and P15 strongly depleted in vivo; P11 strongly enriched in vivo (Fig. 4d); and P2 the most plastic state in normoxia and hypoxia (Figs. 2h and 3f), partially depleted in vivo. While all subpopulations formed tumors, we observed a difference in mouse survival (Fig. 4f). Only mice bearing P2 developed tumors as fast as the parental cells, whereas P6, P11, and P15 grew slower. Interestingly, all subpopulations changed phenotype in vivo resembling the original in vivo equilibrium (Fig. 4g, Supplementary Data 1H). This shows that the subpopulations were able to undergo state transitions in vivo toward an in vivo environment-specific equilibrium. Differences

in tumor development in vivo are linked to the time required by each subpopulation to reach the final environment-specific heterogeneity. This suggests that high tumorigenic potential may be linked to the fast adapting cells with enhanced intra-tumoral phenotypic heterogeneity. Our data further support recent findings[27], suggesting a key role of intra-tumoral heterogeneity and reciprocal crosstalk of phenotypically different tumor cells in creating tumor growth promoting niches.

**Phenotypic states display a similar transcriptome.** To correlate intra-tumoral heterogeneity at the phenotypic and transcriptomic level we applied single cell sequencing using the Drop-seq method[46] on three PDOXs and two GBM cultures. Although a certain degree of intra-tumoral heterogeneity was observed within

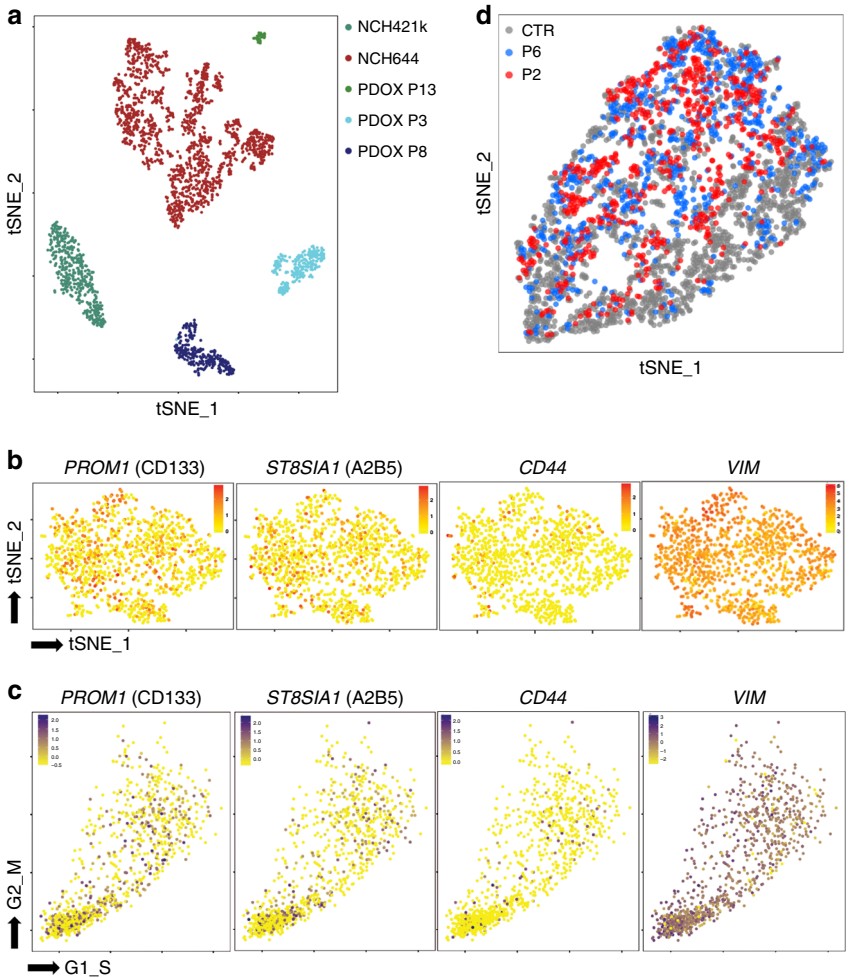

**Fig. 5** Distinct phenotypic states carry similar transcriptome. **a** Overall gene expression relationship between single cells of three GBM PDOXs and two GBM cultures. Patient-derived cells are color coded. **b** Expression of marker genes in NCH644 (expression gradient color coded). See Supplementary Fig. 7 for more examples. **c** Estimation of cell cycle state of individual cells on the basis of relative expression of G1/S and G2/M gene sets. **d** Gene expression relationship between subpopulations P2 and P6 and the original heterogeneous GBM cultures (Control = CTR). Each sample is color coded. See Supplementary Fig. 7C, D for further analysis

each tumor, we did not detect distinct sub-clusters and each tumor showed similar transcriptional cell-to-cell variability (Fig. 5a, cell-to-cell correlation coefficients: 0.72–0.86). If detected, transcripts coding for cell membrane markers were uniformly distributed across the tumor (Fig. 5b, Supplementary Fig. 7A) and their expression was not linked to the cell cycle (Fig. 5c). This was also true for intracellular stemness (*NES, VIM*) and differentiation markers (*GFAP*) (Supplementary Fig. 7A).

As transcripts coding for cell membrane markers were generally detected at low levels (Supplementary Fig. 7B), we were not able to discriminate 16 phenotypic states based on mRNA expression. Therefore, we performed Drop-seq on two subpopulations: P2 (representing the most adaptive phenotypic state in normoxia, hypoxia and in vivo) and P6 (representing cells positive for all stem cell-associated markers but being less adaptive). Interestingly, the two phenotypic states displayed similar transcriptomic profiles to each other and to the heterogeneous parental culture (Fig. 5d, Supplementary Fig. 7C, D), suggesting no profound transcriptomic differences between subpopulations carrying different levels of plasticity. The lack of distinct cell subpopulations at the transcriptomic level further questions a strong hierarchical organization of subpopulations defined by cell membrane markers and suggests subtler molecular

differences between different phenotypic states grown under the same culture conditions.

**Limited impact of chemotherapy on CSC-associated phenotype**. We next asked whether marker expression was affected by treatment. Although GBM resistance to Temozolomide (TMZ), a DNA alkylator, has been suggested to be conferred by CSCs selectively surviving treatment[47], conflicting data exist with regard to CSC chemoresistance in GBM[48,49]. In order to mimic clinically relevant drug doses (5–50 μM) we applied a low TMZ concentration (i.e., IC20 of corresponding cell line). This only led to minor changes of subpopulations in NCH644 TMZ-resistant cells even after long-term treatment (Fig. 6a, Supplementary Data 1I). Minor changes were also observed in NCH421k TMZ-sensitive cultures, with an increase in heterogeneity at day 7 (Fig. 6a, Supplementary Data 1J). Similarly to hypoxia, the changes in time were not always gradual suggesting adaptation via intermediate states. Since a sublethal dose was applied, cell death could be ruled out as a significant factor in phenotypic selection. Furthermore, no major changes were detected at the transcript level after short-term exposure (24 h) to high dose of TMZ (Supplementary Table 2), nor did we observe phenotypic

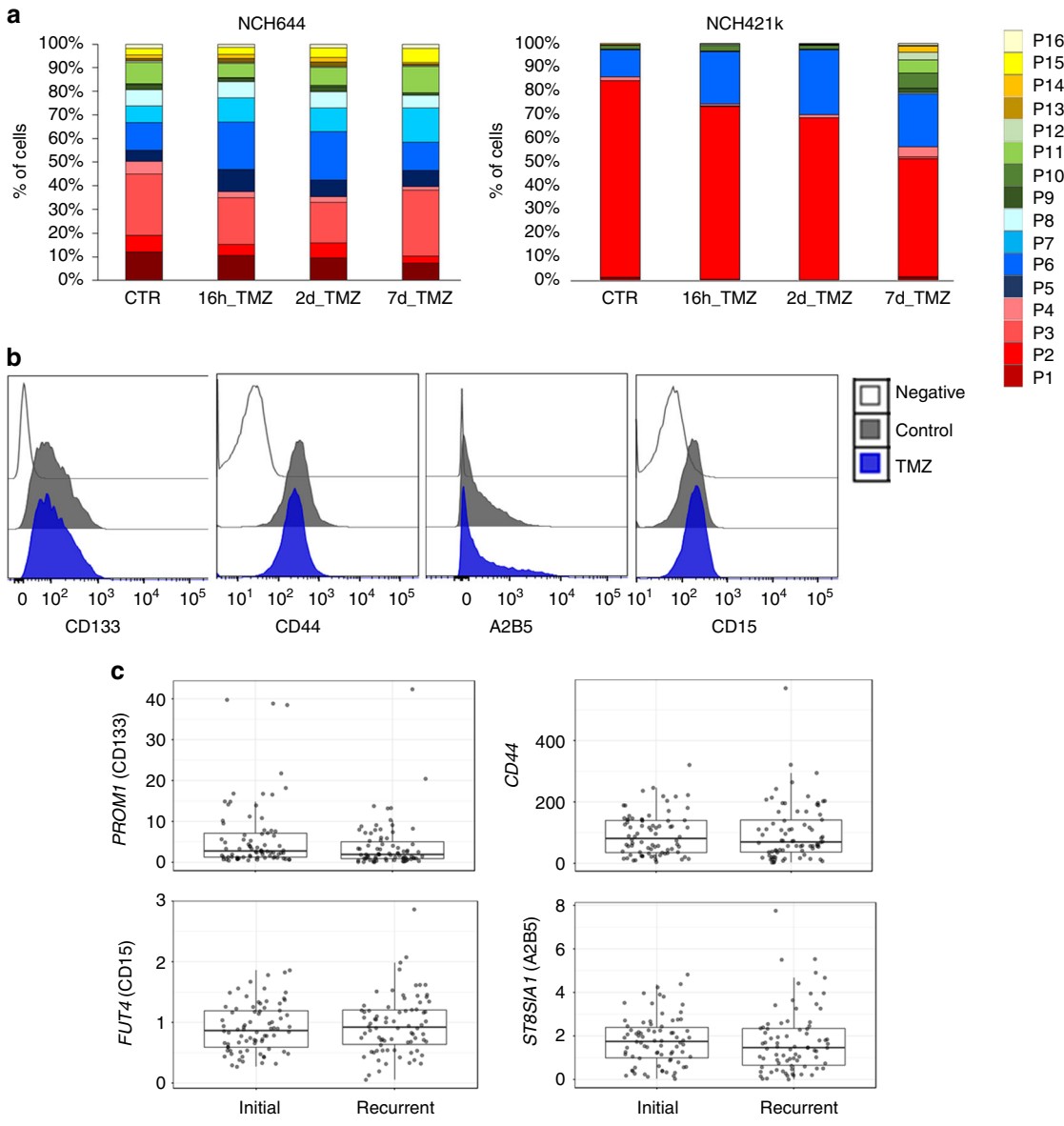

**Fig. 6** Phenotypic heterogeneity in GBM after treatment. **a** Distribution of subpopulations upon TMZ treatment. Cultures were treated with TMZ for 16 h, 2 days, and 7 days. See Supplementary Data 1I, J for statistics. **b** Phenotyping of tumor cells revealed no changes of marker expression upon TMZ treatment in vivo (PDOX T16). **c** Comparative gene expression analysis of genes coding for four cell membrane markers in paired primary and recurrent IDHwt GBM samples ($n = 78$) based on RNA-seq data from the RecuR cohort[33]. Box limits indicate the 25th and 75th percentiles and center lines show the medians as determined by R software; whiskers represent the extreme low and high observed values, unless those are above 1.5 times interquartile range (IQR)—thereby whiskers are limited to 1.5 IQR. All outlying data points are represented by dots. No significant statistical differences were detected (two sided paired $t$-test)

changes in GBM upon long term TMZ treatment in vivo (Fig. 6b). This was confirmed in paired patient samples of recurrent GBMs, where no transcriptomic differences in marker expression were observed after treatment (Fig. 6c). Similarly, we did not detect changes in marker expression following anti-angiogenic treatment with bevacizumab, an antibody against vascular endothelial growth factor (VEGF) (Supplementary Fig. 6H)[50]. Overall, these data indicate that subpopulations of cells expressing CSC-associated markers are not selectively enriched by current treatment approaches, questioning their relevance as a resistance mechanism.

## Discussion
Functional heterogeneity of cancer cells is determined not only by the genetic makeup but also by non-genetic programs such as

stemness features and interactions with the microenvironment. Although it is widely appreciated that cancer cells with stem cell properties exist within solid tumors, the clinical significance of CSC properties is still an open question. We find that phenotypic heterogeneity based on cell membrane markers in GBM is a dynamic process of reversible state transitions. All GBM subpopulations displayed stem cell properties and were tumorigenic, supporting the notion that cancer cells are highly plastic in their response to microenvironmental cues. Despite the absence of a phenotypic hierarchy, we noted a functional difference reflected in tumor development time in vivo, which appeared to be correlated with the ability to reconstitute phenotypic heterogeneity. These data advise against targeting a small subpopulation of CSCs defined by cell membrane markers and highlight the importance of considering these dynamic processes during treatment design.

Our study is in line with previous reports demonstrating strong inter-patient heterogeneity and the difficulty to identify bona fide CSC markers. Only a subset of markers fulfilled the CSC criteria of heterogeneous expression within tumors. We show that the expression of CSC markers in vivo is not intrinsic to specific genetic clones, but is an adaptation process in response to external cues. In GBM cultures the cell membrane marker-associated phenotype was flexible, all subpopulations carried stemness properties and reconstituted heterogeneity. Importantly, we show that the phenotypic heterogeneity could also be recreated by single cells of different phenotypic profiles. Although lineage tracing is a powerful technique to follow the development of individual clones and their proliferative capacities[5], these techniques are currently not applicable for functional studies in combination with multicolor cell membrane phenotyping, limiting our study to reconstitution of clones from single FACS-sorted cells. We further show that phenotypic heterogeneity is created via stochastic non-hierarchical state transitions between phenotypic states, in line with previous observations where CD133 negative GBM cells were able to revert to the CD133 + phenotype in vivo[14]. None of the phenotypic states was unipotent and irreversible in the tested microenvironments, including normoxia, severe hypoxia and in vivo. GBM cells also reverted from a more differentiated state, in line with the disappointing effects of differentiation therapy[38]. Despite potential differences between solid tumors, our data are in agreement with reports from other cancers, including breast and melanoma[51–53]. While we cannot exclude a partial selection upon environmental pressure, we did not observe the complete eradication of subpopulations under any conditions, including drug treatment. Moreover, all populations adapted toward the most optimal equilibrium, suggesting that state transitions are tightly regulated to reach the best fitted balance. This is in contrast to previous reports suggesting targeted selection of CSCs and the loss of differentiation capacities in hypoxia[42,43]. For example, we find that all subpopulations, regardless of their initial phenotype, increased CD133 expression in hypoxia in a reversible manner, arguing against a targeted selection of CD133+ cells. CD133 was also activated in initially negative cultures (Supplementary Fig. 4E) and also CD133 high hypoxic cells were able to undergo aberrant differentiation. In line with previous reports[54], we observe fast and reversible activation of marker expression at the transcriptomic level under hypoxic conditions. Although further studies are needed to identify the factors leading to phenotypic adaptation in vivo in the spatial and temporal context, stromal cells and the metabolic landscape of the brain are likely to play a key role[1]. Macrophage/microglia[24], pH[26], and glucose levels[25] have been proposed to be implicated in phenotypic shifts, with the two latter parameters being intimately associated with the hypoxic niche. Reversibility to CSC-like states is also possible via activation of defined transcriptional factors[55].

Although all cells carried stem cell properties and were tumorigenic, we observed differences in the speed of plasticity, which correlated to tumor development in vivo. A shorter tumor development time was linked to a higher level of plasticity, reaching a favorable environment-specific heterogeneity more quickly. Our data are in agreement with a recent study, showing that reciprocal crosstalk of GBM cells displaying different phenotypes promotes tumor growth[27]. We propose that an increased tumorigenic potential of CSC-like states may be a consequence of an accelerated adaptation to the microenvironment, rather than a unique multipotency per se. This adds a novel criteria to the CSC-like state and may explain some of the reported controversies in functional assays: while all cells are plastic, some adapt faster to new environments in vivo, leading to differences between in vitro self-renewal and in vivo transplantation assays. As highlighted previously[56], the outcome of these assays strongly depends on experimental conditions (e.g., culture conditions of purified subpopulations and implanted cell number) and endpoint determination (i.e., the time allowed for tumor development in mice).

Why some states appear more plastic than others is currently not clear. In addition to the genetic status, epigenetic flexibility regulating transcriptional networks may play a role[57]. Phenotypically distinct subpopulations retained a largely similar transcriptome, suggesting more subtle differences within GBM compared with, e.g., breast cancer[51,53]. This is in line with recent single cell RNA-seq data, revealing a continuous stem cell signature[4]. It remains to be seen if this is confirmed in other patient-derived cells and if more differences are present between less abundant coding and non-coding RNA transcripts, which are below the sensitivity of the Drop-seq method. More important transcriptomic differences could be expected if tumor cells are acutely extracted from in vivo tumors, where cells may reside in different niches. Although not addressed in this study, the changes of epitope presentation upon environmental impacts can arise from different mechanisms, including mRNA and protein expression changes, changes in protein localization and conformation; and/or epitope posttranslational modifications influencing epitope recognition[17,32].

Although initial reports suggested targeted selection of putative CSCs upon treatment[47,58], later studies could not find evidence of CSC enrichment following chemotherapy[48,49]. In some studies, GBM CSCs were more resistant to radiation compared with their non-CSC conterparts, however, non-CSCs also remained relatively radioresistant[59] and acquired similar properties to CSCs upon exogenous treatment[60]. Here, we did not detect major phenotypic changes neither upon long-term treatment in vitro with clinically relevant TMZ doses, nor upon short-term treatment with a high dose. No major changes were observed also in vivo after treatment with TMZ or bevacizumab, arguing against a selective survival of CSC subpopulations or a role in intrinsic resistance. Although high TMZ doses over a prolonged period of time could potentially lead to a stronger selective pressure and the selection of certain phenotypic states, this condition is unlikely to be achieved in patients with the current treatment protocol (therapeutically achievable dose: 5–50 μM[61]). Of note, we have previously shown that neither TMZ, nor bevacizumab activates the stem cell-associated efflux property, defined as the side population phenotype, in GBM cells[34,62]. In summary, these data highlight the limitations of therapeutic strategies targeting cell membrane marker-defined CSCs, including CSC-directed immunotherapies[63].

Phenotypic plasticity is a powerful mechanism for overcoming selective pressures both during tumor development and adaptation to changing microenvironments, e.g., nutrient supply, oxygen level or treatment. Studing the dynamic feedback of tumor cells and their in vivo niche will require the development of complex mathematical models that explicitly include such interactions. In keeping, advanced mathematical simulations of state transitions following combinatorial treatment predicted the survival of the most plastic clones[64]. In breast cancer, it was shown that the tumorigenic potential was lost by inhibiting the transition from CD44-low toward CD44-high cells[53]. Elucidating state transition programs and mechanisms driving cellular plasticity will be essential to overcome current therapeutic limitations.

## Methods

**TCGA gene expression analysis**. The gene expression pattern across glioma patients was investigated using The Cancer Genome Atlas (TCGA) cohort via GlioVis data portal[65] and Cbioportal[66]. Expression value corresponds to median-centered RNA-seq data, DNA methylation data were from a merged dataset on 25,978 probes shared by the HM27 and HM450 platforms analysis (#syn2486658)),

genomic copy number alterations were from PanCan12 Genom (https://www.synapse.org/). Putative copy number calls on 563 cases determined using GISTIC 2,0–2 = homozygous deletion; −1 = hemizygous deletion, 0 = neutral, 1 = gain, 2 = high level amplification. Mutation data are from whole-exome sequencing of 291 tumors. Heatmaps were generated using Gitools software[67]. Group comparison was performed using Mann–Whitney–Wilcoxon, and Benjamini Hochberg multiple test correction was applied (at $p = 0.05$, Mann–Whitney–Wilcoxon null hypothesis: correlation = 0). Gene expression data analysis of longitudal cohort of IDHwt GBMs[33] was performed through http://recur.bioinfo.cnio.es/ with RNA-seq ($n = 78$ per group) and AffymetrixU133A ($n = 23$ per group) datasets. Statistical analysis was performed by two-sided paired $t$-test.

**Patient-derived orthotopic xenografts (PDOXs)**. GBM samples were collected at Centre Hospitalier in Luxembourg (Neurosurgical Department) or Haukeland University Hospital (Bergen, Norway) from patients having given informed consent, and with approval from the local ethics committees (National Ethics Committee for Research (CNER), Luxembourg, local ethics committee Haukeland University Hospital, Bergen). All biopsies were from IDHwt GBMs. Organotypic GBM spheroids were derived from patient samples and cultured in DMEM medium, 10% FBS, 2 mM L-Glutamine, 0.4 mM NEAA, and 100 U ml⁻¹ Pen-Strep; Lonza in agar pre-coated flasks for 7–10 days. Spheroids were implanted in the brain of eGFP-expressing Nod/Scid mice[34,36]. Animals were sacrificed at the appearance of neurological symptoms and/or weight loss. PDOX models are available from the corresponding author or via EuroPDX consortium (https://www.europdx.eu/) on reasonable request.

The GBM patient-derived xenograft (PDOX T16, $n = 5$) was used for Temozolomide (TMZ) treatment (unmethylated *MGMT* promoter, in vitro IC$_{50}$ for TMZ ≈ 1 mM), receiving intraperitoneal injections of TMZ (Sigma 200 mg kg⁻¹ in 10% DMSO) three times a week, starting 2 weeks after spheroid implantation. For anti-angiogenic treatment, mice (PDOX P3, $n = 5$) received weekly intraperitoneal injections of bevacizumab (Avastin, Roche; 20 mg kg⁻¹ in saline), starting 3 weeks after spheroid implantation[50]. Control animals received injections of 10% DMSO or saline, respectively. The handling of the animals and the surgical procedures were performed in accordance with the regulations of the European Directive on animal experimentation (2010/63/EU), i.e., the experimental protocols were approved by the local ethics committee (Animal Welfare Structure of the Luxembourg Institute of Health) and by the Luxembourg Ministries of Agriculture and of Health.

**Glioblastoma patient-derived cultures**. The GBM serum-free cultures NCH421k, NCH660h, NCH465, NCH601, and NCH644, were provided by Dr Christel Herold-Mende (Department of Neurosurgery, University of Heidelberg)[35]. GBM cells were derived immediately after dissociation of primary patient tumor and culture in 3D serum-free conditions. NCH421k, NCH660h, NCH465, and NCH601 were cultured as non-adherent spheres in DMEM-F12 medium (Lonza) containing 1× BIT100 (Provitro), 2 mM L-Glutamine, 30 U ml⁻¹ Pen-Step, 1 U ml⁻¹ Heparin (Sigma), 20 ng ml⁻¹ bFGF (Miltenyi, 130-093-841), and 20 ng ml⁻¹ EGF (Provitro, 1325950500). NCH644 grew in Neurobasal® base medium (Life Technologies) supplemented with 1× B27 (Life Technologies) 2 mM L-Glutamine, 30 U ml⁻¹ Pen-Step, 1 U ml⁻¹ Heparin (Sigma), 20 ng ml⁻¹ bFGF (Miltenyi, 130-093-841), and 20 ng ml⁻¹ EGF (Provitro, 1325950500). The GBM cultures TB101 and TB107, kindly provided by Dr. Håkan Hedman (Umeå University, Sweden), were cultured in DMEM-F12 medium (Lonza) containing 1× B27 and 1× N2 supplements (Provitro), 2 mM L-Glutamine, 30 U ml⁻¹ Pen-Step, 1 U ml⁻¹ Heparin (Sigma), 20 ng ml⁻¹ bFGF (Miltenyi, 130-093-841), and 20 ng ml⁻¹ EGF (Provitro, 1325950500). P13NS short term cultures were derived from PDOX P13 and cultured in serum-free medium based on Neurobasal® base medium as described above. U87 cells (obtained from ATCC, HTB-14) were cultured as monolayers in DMEM containing 10% FBS, 2 mM L-Glutamine, and 100 U ml⁻¹ Pen-Strep (Lonza). Classical normoxic cultures were performed at 37 °C under 5% CO$_2$ atmospheric oxygen. pH of the cultures was monitored with phenol red. GBM cultures were tested for cell membrane marker expression profiles over several passages with no major changes observed over time. Similarly no difference in marker distribution was observed within the individual 3D spheres of different sizes (average sphere size 300 μm). Small changes in the composition of serum-free media had no impact on the cell membrane marker profiles. Cell lines were regularly tested for mycoplasma contamination. Cell lines were authenticated by DNA profiling using SNP-based multiplex approach and compared with the other continuous cell lines in the DSMZ database. SNP profiles were unique.

**Hypoxia, differentiation, and TMZ treatment**. In vitro TMZ (Sigma) treatment was performed at sublethal doses (200 μM for NCH644 and 25 μM for NCH421k) for indicated time points (16 h, 2 days, and 7 days). NCH421k has a methylated *MGMT* promoter and a low IC$_{50}$ (272 μM). NCH644 have an unmethylated *MGMT* promoter (IC$_{50}$ ≈ 227 mM). The experiment was repeated independently four times ($n_B = 4$) with three technical replicates per subpopulation ($n_T = 3$). TMZ was refreshed every 3 days. During hypoxia cells were maintained at 0.5% O$_2$ in a hypoxic incubator (Galaxy 48R incubator, New Brunswick) for the indicated time points (16 h, 48 h, 7 days, and 60 days). Experiments were repeated

independently three times ($n_B = 3$) with three technical replicates per subpopulation ($n_T = 3$). To avoid acidification of the hypoxic cultures, the medium was regularly replaced with fresh media pre-equilibrated in hypoxia. For differentiation, cells were grown as adherent cultures in 10× diluted Matrigel in original medium depleted from bFGF and EGF and supplemented with 10% FBS and 10 ng ml⁻¹ All-Trans Retinoic acid (ATRA, Sigma) for 14 days. Cells were re-adapted to normoxic 3D sphere cultures for an additional 14 days in original culture media. Experiments were repeated independently three times ($n_B = 3$) with three technical replicates per subpopulation ($n_T = 3$).

**Immunostaining**. NCH644 spheres were embedded in Tissue-Tek® O.C.T.™ Compound (Sakura Finetek) and flashfrozen at −80 °C. Immunocytochemistry was performed on 10 μm sections, fixed in 4% PFA for 10 min. Adherent cultures were fixed directly in the plates. Fixed samples were blocked for 30 min in TBS supplied with 0.1% Triton-X100 and 2% FBS and incubated with primary antibodies (see Supplementary Table 3 for antibodies used) and 1 μg ml⁻¹ DAPI (Invitrogen). Secondary antibody staining was performed when needed. For image acquisition, a Leica® DMI6000 B inverted microscope with a Leica® 350 FX camera was used with its concordant Leica Application Suite® software.

**Flow cytometry settings**. Data acquisition was performed on a FACS Aria™ SORP cytometer (BD Biosciences) fitted with a 640 nm (30 mW) red laser, a 355 nm (60 mW) UV laser, a 405 nm (50 mW) violet laser, a 488 nm (100 mW) blue laser, and a 561 nm (50 mW) yellow/green laser. The Hoechst dye was excited by the UV laser and fluorescence was collected in two channels: UV-1 450/50 band-pass (BP) filter and UV-2′ 660/40 long-pass (LP) filter. An LP 635 nm dichroic mirror was used to split the emission wavelengths. The instrument was calibrated each time with Cytometer Setup&Tracking Beads (BD Bioscience). The Coefficient of Variation of the instrument (% CV) was routinely examined before each experiment. A 100 μm nozzle and window extension (WE) 3 were used for data acquisition and sorting. Imaging flow cytometry was performed with an Image-Stream imaging cytometer (Amnis) fitted with a 375 UV laser, a 488 blue laser, a 561 yellow-green laser, a 642 red laser, and a 785 nm infrared laser. Acquisition was performed with the INSPIRE® software and analysis was performed using IDEAS® image analysis software. Pictures were taken at ×60 magnification at low-speed high-sensitivity mode.

**Multicolor cell membrane phenotyping**. Cell cultures were dissociated using Accutase® (Sigma-Aldrich). Xenografts were dissociated with MACS Neural Tissue Dissociation Kit (P) (Miltenyi) following the manufacturers' instructions. Single cells were resuspended in HBSS, 2% FBS, 10 mM HEPES buffer (100 μl per test). Cells were incubated with the IR-LIVE/DEAD® Fixable Dead Cell Stains (Invitrogen; 1 μg ml⁻¹) and appropriate preconjugated antibodies for 30 min at 4 °C in the dark (Supplementary Table 3). For cell cycle analysis in viable cells, cells were prestained with Hoechst 33342 (5 μg ml⁻¹, Bisbenzimide, Ho342; Sigma) at 37 °C before antibody staining[37]. Data acquisition was performed on a FACS Aria™ SORP cytometer (BD Biosciences) and ImageStream imaging cytometer (Amnis). Data acquisition and analysis were done for FACSAria with DIVA software (BD Bioscience); and INSPIRE and IDEAS® for ImageStream. Histograms were prepared with the FlowJo software.

**Intracellular marker phenotyping and apoptosis test**. Cell cultures were dissociated using Accutase® (Sigma-Aldrich) Xenografts were dissociated with MACS Neural Tissue Dissociation Kit (P) (Miltenyi) following the manufacturers' instructions. Single cells were resuspended in HBSS, 2% FBS, 10 mM HEPES buffer (100 μl per test). Cells were incubated with the IR-LIVE/DEAD® Fixable Dead Cell Stains (Invitrogen; 1 μg/ml) and appropriate preconjugated antibodies for 30 min at 4 °C in the dark (Supplementary Table 3). Cells were fixed with the BD Cytofix™ solution for 20 min and permeabilised in the BD Perm/Wash™ for 10 min at RT. Cells were incubated with appropriate preconjugated antibodies for 30 min at 4 °C in the dark (Supplementary Table 3). Data acquisition was performed on a FACS Aria™ SORP cytometer (BD Biosciences). For apoptosis test cells were resuspended in 100 μl of binding buffer (HBSS w/o Ca 2 + /Mg2 + (Sigma-Aldrich), 2% FBS and 0.01 M Hepes pH 7,4, 0.14 M NaCl (Sigma-Aldrich), and 2.5 mM CaCl₂ (Sigma-Aldrich)). Staining with AnnexinV-APC (Immunotools) was performed for 30 min at room temperature in dark. Cells were subsequently washed with binding buffer and kept on ice until analysis on flow cytometer. In total, 50 μg ml⁻¹ propidium iodide (PI) (Invitrogen) was added 5 min before data acquisition. The experiment was performed in three biological replicates ($n_B = 3$) with three technical replicates each ($n_T = 3$). Statistical differences were calculated using Student $t$-test with Bonferroni multiple-significance-test correction for four comparisons.

**Self-renewal test in vitro**. Single cells from 16 subpopulations were FACS-sorted to a 96-well plate (1 cell per well; one 96-well plate per subpopulation) and cultured for 4 weeks in normoxia or 0.5% O$_2$ hypoxia. Spheres derived from each subpopulation were collected (passage 1) and single viable cells were resorted and plated as single cells (one 96-well plate per subpopulation, 1 cell per well). Replating was done three times to reach four passages in total. Only spheres > 40 μm were considered as a positive result for sphere forming capacity. Total

sphere number and average sphere size ($n = 20$ per subpopulation if available) were recorded at each passage before cell harvesting. Each subpopulation at each passage was phenotyped as described above. FACS-sorted viable single cells from the bulk cells were used as control. The experiment was repeated independently four times. Significant differences in sphere number across populations and passages were evaluated with the Kruskal–Wallis test. Significant differences in sphere size were tested with mixed linear models with either subpopulation or passage as fixed-effects and considering batch effect as random.

**Proliferation and mulitpotency test.** In total, 300 cells of each NCH644 subpopulation were FACS-sorted to a 48-well plate and cultured for 20, 30, and 70 days in normoxia or 0.5% $O_2$ hypoxia. At each time point, cells derived from each subpopulation were phenotyped as described above. Total cell number was recorded with the Countess® cell counter (Invitrogen) after 20 and 30 days of culture to determine the proliferation rate. Doubling time was calculated as follows: doubling time $= \frac{t_2 - t_1}{\log_2 \frac{\text{cell number}_2}{\text{cell number}_1}}$, where $t_1$ and $t_2$ represent time points. FACS-sorted single viable cells from bulk were used as control. The experiment was repeated independently four times ($n_B = 4$) with three technical replicates per subpopulation ($n_T = 3$). Significant differences of doubling times were tested with mixed linear models with subpopulation as fixed-effects and considering plate effects as random. The proportion of each subpopulation was calculated as the percentage of viable single cells. The column chart graphs show mean percentage of technical and biological replicates. Error bars were omitted for visualization purposes. Alluvial plots have been generated under R using the package *alluvial* available at https://github.com/mbojan/alluvial. Significant differences between phenotypic states were calculated with the Student's *t*-test with Bonferroni's multiple comparison test/correction.

**Mathematical modeling.** Markov model principles: To quantify the transitions between the 16 phenotypes we applied Markov chain modeling implemented in the freely available R package *CellTrans* (http://github.com/tbuder/CellTrans)[41]. The model is based on the assumptions that cell state alterations occur due to stochastic cell state transitions only depending on the current state of the cell and possibly the experimental environment (e.g., hypoxia) and that proliferation rates of the involved phenotypes are approximately equal. Since all examined subpopulations proliferated at a similar rate both in normoxia (Fig. 2c) and hypoxia (Fig. 3c), the proliferation rates were neglected for the estimation of transition rates. In vitro experiments were performed under two different environmental conditions (normoxia, hypoxia) and transition rates were estimated separately for each condition. In the absence of known data about inter-cellular interactions, these components were omitted. Our modeling approach led to a Markov chain with a transition matrix containing the probabilities of state transitions allowing to discriminate frequent and non-frequent state transitions and identify hierarchical or non-hierarchical transition behavior. If the underlying network is irreducible, each state can transit directly or via intermediate steps into any other state. This behavior implies that stochastic state transitions are non-hierarchical and therefore reversible in a biological sense. In contrast, if the transition network has a tree structure, it corresponds to a perfect hierarchy. Intermediate network structures are possible as well, which imply some degree of hierarchy between transient states at the top of the hierarchy and recurrent states at the bottom of the hierarchy. Moreover, on the basis of the estimated transition matrix, it is possible to predict the composition of the population in equilibrium by calculating the Markov chain stationary state. In addition, the time from a specific initial composition until an approximate equilibrium is reached can be estimated.

Construction of data matrices: Let $w_{i,j}^{(t)}$ denote the experimentally observed mean proportion of phenotype $j$, $j = 1,\dots,16$, in the experiment starting with pure subpopulations of phenotype $i$, $i = 1,\dots,16$, at time $t$. Then, one can construct a phenotype proportion matrix at time $t$ as follows:

$$W^{(t)} = \begin{pmatrix} w_{1,1}^{(t)} & \cdots & w_{1,16}^{(t)} \\ \vdots & \ddots & \vdots \\ w_{16,1}^{(t)} & \cdots & w_{16,16}^{(t)} \end{pmatrix}$$

Note that the phenotype proportion matrix describing the initial proportions is the $16 \times 16$ identity matrix, i.e.

$$W^{(0)} = \begin{pmatrix} 1 & 0 & \cdots & \cdots & 0 \\ 0 & 1 & 0 & \cdots & 0 \\ \vdots & \ddots & \ddots & \ddots & \vdots \\ \vdots & \ddots & \ddots & \ddots & 0 \\ 0 & 0 & \cdots & 0 & 1 \end{pmatrix}$$

The transitions between the phenotypes were estimated as probabilities of state transition per time-step of the underlying Markov chain. Importantly, the implications of the model with respect to hierarchical structure, equilibrium composition and relaxation time are highly independent of the choice of the time-step length. Here, we choose a time-step length of 1 day. We obtained three phenotype matrices from the experimental data in normoxia: $W^{(20)}$ after 20 days,

$W^{(30)}$ after 30 days, and $W^{(70)}$ after 70 days. In hypoxia we constructed only one matrix, namely $W^{(60)}$ for the measurement after 60 days.

Estimation of the transition probabilities: Markov chain theory allows to connect the initial phenotype proportion matrix with the phenotype matrix after $n$ time-steps with the equation $W^{(0)}P^n = W^{(n)}$. This equation can be solved for the underlying transition matrix $P$, i.e.

$$\hat{P}_{(n)} = \left( \left( W^{(0)} \right)^{-1} W^{(n)} \right)^{\frac{1}{n}}.$$

Hence, we obtained the three transition matrices $\hat{P}_{(20)}$, $\hat{P}_{(30)}$, and $\hat{P}_{(70)}$ in normoxia which needed to be regularized in order to obtain stochastic matrices, see ref. [41] for details. These matrices were averaged to obtain a final estimate for the transition matrix $P$, i.e., $\hat{P} = \frac{\hat{P}_{(7)} + \hat{P}_{(10)} + \hat{P}_{(23)}}{3}$. In hypoxia, $P = \hat{P}_{(60)}$ was used since there was only one time point of measurement.

Krackhardt hierarchy: To estimate the degree of hierarchy of the estimated transition matrixes from data obtained under different environmental conditions, graph hierarchy introduced by Krackhardt was calculated[68]. The degree of deviation from pure hierarchy, i.e., tree structure, is assessed by counting the number of pairs that have reciprocated ties relative to the number of pairs where there is not any tie, i.e., the proportion $p$ of all tied pairs having reciprocated ties. Krackhardt hierarchy is then defined as $1-p$. A perfect hierarchy is characterized by no reciprocated ties and exhibits a Krackhardt hierarchy of one. Calculations were performed in R using hierarchy function with Krackhardt measure.

Marker dependency calculation: We investigated whether the cell state transitions between positive and low/negative expression of one marker in normoxia and hypoxia were dependent on the level of expression of the other markers. For each marker, we distinguished two phenotypic states with respect to positive and low/negative expression of this marker. In detail, we added the percentage of all CSC-associated subpopulations with positive or low/negative expression of that marker in the measured plasticity data for each time point of measurement. This resulted in $2 \times 2$ cell state proportion matrixes per time point. This coarse-grained data could be utilized to derive corresponding $2 \times 2$ cell state transition matrixes and corresponding steady states by applying CellTrans[41]. The steady states of these $2 \times 2$ cell state transition matrices could then be compared with the fractions of the corresponding phenotype when considering all four markers in the analysis. Therefore, percentage of subpopulations with the same expression of a single marker were added to obtain the steady state proportion for each individual marker. Nearly equal equilibrium proportions indicate independent transitions whereas large differences indicate a dependency. Calculations were performed for normoxic and hypoxic datasets.

**Cell line-derived xenografts and in vivo tumor formation.** GBM patient-derived cultures were implanted intracranially to NOD/Scid mice (50,000 per mouse for NCH644 and NCH421k, 300,000 cells for NCH660h and NCH601, $n = 3$) into the right frontal cortex using a stereotactic device. Animals were sacrificed at the appearance of neurological symptoms and weight loss. To test tumorigenicity in vivo, 5000 cells of P2, P6, P11, and P15 subpopulations and original NCH644 cells were engrafted into nude mice directly after sort ($n = 7$ per group). Animals were monitored daily and the following criteria were evaluated: (1) loss of more than 10% of body weight, (2) exhibition of strong neurological signs (difficulty ambulating or abnormal movement), (3) increased lordosis, or (4) swollen belly. The criteria were scored as follows: 0 = none, 1 = early, 2 = established, 3 = severe signs and animals were sacrificed when three criteria with grade 2 or one criterion with grade 3 were reached. Kaplan–Meier survival curves, log-rank test for survival analysis and IC50 were calculated with the GraphPad Prism5. Tumor developed in xenografted mice were FACS phenotyped as described above.

**Single cell RNA-Seq using Drop-Seq.** To obtain a pure population of single viable cells all GBM cultures were FACS-sorted (NCH644, NCH421k bulk cultures and NCH644 subpopulations P2 and P6). For GBM PDOXs we have pre-selected hCD90 positive tumor cells. FACS-sorted populations were collected in HBSS, 0.5% BSA at 4 °C. Prior to cell loading on the Drop-seq chips, the viability of cells was verified and concentration was adjusted to ~150 cells μl$^{-1}$ as optimal concentration to achieve single-cell encapsulation within each droplet of ~ 1 nl. All samples analyzed had a cell viability > 95%.

Microfluidics devices were fabricated using a previously published design[46]. Softlithography was performed using SU-8 2050 photoresist (MicroChem) on 4″ silicon substrate to obtain a feature aspect depth of 100 μm. After overnight silanization (using Chlorotrimethylsilane, Sigma), the wafer masks were used for microfluidics fabrication. Drop-seq chips were fabricated using silicon based polymerization chemistry. Briefly, polydimethylsiloxane (PDMS) base and cross-linker (Dow Corning) were mixed at a 10:1 ratio, mixed and degassed before pouring onto the Drop-seq master template. PDMS was cured on the master template, at 80 °C for 2 h. After incubation and cooling, PDMS slabs were cut and the inlet/outlet ports were punched with 1.25-mm biopsy punchers (World Precision Instruments). The PDMS monolith was plasma-bonded to a clean microscopic glass slide using a Harrick plasma cleaner. Immediately after pairing the plasma-treated surfaces of the PDMS monolith and the glass slide, flow channels of the Drop-seq chip were subjected to a hydrophobicity treatment using

1H,1H,2H,2H-Perfluorodecyltrichlorosilane (in 2% v/v in FC-40 oil; Alfa Aeser/Sigma). After 5 min of treatment, excessive silane was blown through the inlet/outlet ports. Chips were further incubated at 80 °C for 15 min.

Experiments followed the original Drop-seq protocol[46] with minor changes. Synthesized barcoded beads (Chemgenes Corp., USA) were co-encapsulated with cells inside the droplets containing lysis reagents using an optimal bead concentration of 200 beads μl$^{-1}$ in Drop-seq Lysis buffer medium. Cellular mRNA was captured on the beads via barcoded oligo (dT) handles synthesized on the surface. For cell encapsulation, 2 ml of cell and bead suspensions were loaded into 3 ml syringes (BD), respectively. To keep beads in homogenous suspension a micro-stirrer was used (VP scientific). The QX 200 carrier oil (Bio-Rad) used as continuous phase in the droplet generation was loaded in a 20 ml syringe (BD). For droplet generation, 3.6 and 13 ml h$^{-1}$ were used in KD scientific Legato syringe pumps for the dispersed and continuous phase flows, respectively. After stabilization of droplet formation, the droplet suspension was collected into a 50 ml Falcon tube. Collection of the emulsion was carried out until 1 μl of the single cell suspension was dispensed. Droplet consistency and stability were evaluated by bright-field microscopy using INCYTO C-Chip Disposable Hemacytometer (Fisher Scientific). Bead occupancy within droplets was carefully monitored to avoid multiple bead occupancy. The subsequent steps of droplet breakage, bead harvesting, reverse transcription, and exonuclease treatment were carried out in accordance to[46]. RT buffer contained 1× Maxima RT buffer, 4% Ficoll PM-400 (Sigma), 1 μM dNTPs (ThermoScientific), 1 U ml$^{-1}$ Rnase Inhibitor (Lucigen), 2.5 μM Template Switch Oligo, and 10 U ml$^{-1}$ Maxima H-RT (ThermoScientific). Post Exo-I treatment, the bead counts were estimated using INCYTO C-Chip Disposable Hemacytometer, and 10,000 beads were aliquoted in 0.2 ml Eppendorf PCR tubes. PCR mix was dispensed in a volume of 50 μl using 1× Hifi HotStart Readymix (Kapa Biosystems) and 0.8 mM Template-Switch-PCR primer. The thermocycling program for the PCR amplification was modified for the final PCR cycles by 95 °C (3 min), four cycles of 98 °C (20 s), 65 °C (45 s), 72 °C (3 min), 10 cycles of 98 °C (20 s), 67 °C (20 s), 72 °C (3 min) and followed by a final extension step of 72 °C for 5 min. Post PCR amplification, libraries were purified with 0.6× Agencourt AMPure XP beads (Beckman Coulter), in accordance with the manufacturer's protocol. Finally, the purified libraries were eluted in 20 μl RNAase/DNAase-free molecular grade water. Quality and concentration of the sequencing libraries were assessed using BioAnalyzer High Sensitivity Chip (Agilent Technologies).

The 3′ end enriched cDNA libraries were prepared by tagmentation reaction of 600 pg cDNA library using the standard Nextera XT tagmentation kit (Illumina). Reactions were performed according to the manufacturer's instructions, except for the 400 nM primer sets replaced by Primer 1 (AATGATACGGCGACCACCGAG ATCTACACGCCTGTCCGCGG AAGCAGTGGTA TCAACGCAGAG T*A*C) and Primer 2 (N703: CAAGCAGAAGACGGCATACGAGA TTTCTGCCTGTCT CGTGGGCTCGG for the NCH644 subpopulation 6 and N709: CAAGCAGAAGA CGGCATACGAGATAGCGTAGCGTCTCGTGGGCTCGG for NCH644 subpopulation 2, NCH644 and NCH421k). The PCR amplification cycling program used was 95 °C 30 s; 14 cycles of 95 °C (10 s), 55 °C (30 s), 72 °C (30 s) followed by a final extension step of 72 °C (5 min). Libraries were purified twice to reduce primers and short DNA fragments with 0.6× and 1× Agencourt AMPure XP beads (Beckman Coulter), respectively, in accordance with the manufacturer's protocol. Finally, purified libraries were eluted in 15 μl molecular grade water. Quality and quantity of the tagmented cDNA library was evaluated using BioAnalyzer High Sensitivity DNA Chip. The average size of the tagmented libraries prior to sequencing was between 400 and 700 bps.

Purified Drop-seq cDNA libraries were sequenced using Illumina NextSeq 500 with the recommended sequencing protocol except for 6 pM of custom primer (GCCTGTCCGCGGAAGCAGTGGTATCAACGCAGAGTAC) applied for priming of read 1. Paired end sequencing was performed with the read 1 of 20 bases (covering the random cell barcode 1–12 bases) and the rest 13–20 bases of random unique molecular identifier (UMI) and for read 2 the 50 bases of the genes.

The FASTQ files were assembled from the raw BCL files using Illumina's bcl2fastq converter and ran through the FASTQC codes [Babraham bioinformatics; https://www.bioinformatics.babraham.ac.uk/projects/fastqc/] to check for consistency in library qualities. The monitored quality assessment parameters were (a) quality per base sequence (especially for the read 2 of the gene), (b) per base N content, (c) per base sequence content, and (d) overrepresented sequences. Libraries that showed significant deviation were re-sequenced. The FASTQ files were then merged and converted to binaries using PICARD's fastqtosam algorithm. The sequencing reads were converted to a digital gene expression matrix (DGE) using the Drop-seq bioinformatics pipeline[46]. To normalize for the transcript loading between the beads, the averaged normalized expression levels (log$_2$(TPM +1)) were calculated. Beads without cellular mRNAs were identified by using a cumulative function of the total number of transcripts per barcode and empirical thresholding on the resulting "knee plot". To filter poor quality reads and cells with low transcript content, only cells with at least 1500 expressed genes and genes at least expressed in 20 cells were considered for further analysis. The average number of UMI-collapsed transcripts per cell was 5970 corresponding to the 2430 genes detected per cell on average.

To remove batch effect, we used independent component analysis ICA (R fastICA package) decomposing the original expression matrix into a product of statistically independent signals and weight matrices: $X_{nm} = S_{nk} \times M_{km}$, where $X_{nm}$ is the log-transformed count matrix for $n$ genes and $m$ cells, $S_{nk}$ is a matrix of $k$ independent components and $M_{km}$ is the weight matrix for each component over $m$ cells. Stability of the ICA decomposing was tested by 100 runs of ICA. The log-transformed count matrix was decomposed using eight independent components. The optimal number of components was selected by minimizing the correlation between rows of weight matrix. We observed that the weight of component #7 was strongly linked to the experimental batch. In accordance with[69], this component was suppressed by setting its weight ($M_{7,i}$, where $i = 1.. m$) to 0, and the normalized data were recovered by matrix multiplication of the components by their weights.

Estimation of the highly variable genes and principal component reduction and tSNE dimensionality reduction was implemented using SEURAT R package (http://satijalab.org/seurat/) or the R package Rtsne with an initial PCA, a perplexity of 40 and a learning rate of 200 (5000 iterations) (https://github.com/jkrijthe/Rtsne). For reproducibility with the original algorithm, the theta value was set to zero.

The correlation coefficient was calculated between each cell after filtering using the Pearson method; the mean value is represented for each group/subpopulation/fraction. Differential expression analysis was performed using DESeq2 package of R. Here the raw integer counts were used for consistency with the algorithm requirements. Centering of the gene expression value was performed by obtaining the relative expression levels, by subtracting the average expression value (log$_2$(TPM+1)) of each gene from all the cells of the gene expression matrix. For cell cycle analysis we have applied two prominent gene expression programs of the G1/S (100 genes) and G2/M phases (133 genes), shown to overlap in the two programs[46]. Due to the sparsity of the single-cell RNA-seq data, the expression data for each cell cycle phase were refined by evaluating the correlation data between each of the genes in the scRNAseq data with the average gene expression values of all the genes involved in the respective cell cycle program (G1/S & G2/M), and including all the genes with high correlation value ($R^2 > 0.3$; $P_{val} < 0.05$, Spearman's correlation). The biaxial plot of G1/S and G2/M programs illustrated in the Fig. 5c is the average score of all the genes involved in the respective cell cycle programs. Figure 5c further represents the expression value of different genes of interest by mapping the expression value onto the respective cells.

**Genome-wide microarray expression analysis.** Gene expression profiles of GBM cells were analyzed after 16 h and 7 days of culture in severe hypoxia (0.1–0.5% O$_2$) and compared with normoxic cells[44,45]. For TMZ treated cells, a high-dose (500 μM) short-term (24 h) treatment was applied. Total RNA was extracted using QIAGEN® RNeasy Mini Kit (Qiagen) or Trizol, according to the manufacturer's protocols and processed (250 ng) using the Affymetrix WT Expression kit before being hybridized to GeneChip® Human Gene 1.0ST Arrays or GeneChip® Human Transcriptome Arrays 2.0 (Affymetrix) (protocol P/N 702808 Rev.6). CEL files containing hybridization raw signal intensities were imported into the Partek GS software for further statistical analysis. R statistical environment was used for hierarchical clustering, principal component analysis and for empirical Bayesian statistics (LIMMA, R/Bioconductor).

**Statistical tests.** The statistical approaches have been chosen based on the data type and measurements across the manuscript. Statistical tests are described in each paragraph above corresponding to the associated experimental procedures.

**Reporting summary.** Further information on research design is available in the Nature Research Reporting Summary linked to this article.

## Data availability

Gene expression profiles across glioma patients were investigated using publicly available datasets: (i) GlioVis data portal[65] http://gliovis.bioinfo.cnio.es/; (ii) Cbioportal[66] http://www.cbioportal.org/; and (iii) RecurR http://recur.bioinfo.cnio.es/. DNA microarray data are available in the ArrayExpress database (https://www.ebi.ac.uk/arrayexpress/) under the accession number E-MTAB-3085. The scRNA-seq data are available in the Gene Expression Omnibus (https://www.ncbi.nlm.nih.gov/geo) under the accession number GSE128195. Scripts supporting mathematical modeling are available in the R package CellTrans (http://github.com/tbuder/CellTrans)[41]. Remaining datasets supporting the findings are available from the corresponding author on reasonable request.

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

## Acknowledgements

We thank A. Oudin, V. Baus, and N. Nicot for technical assistance, H. Hedman (Department of Radiation Sciences, Umeå University, Sweden) for providing glioma cultures and C. Guérin for assistance in the ImageStream experiments (Amnis, Inserm U970, Flow Cytometry Core Facility, France). We are grateful to S. Senn (Competence Centre for Methodology and Statistics, LIH) and T. Bastogne (cINRIA-BIGS & Centre de Recherche en Automatique de Nancy, France) for advice in statistics and mathematical modeling, and to the Clinical and Epidemiological Investigation Center of the LIH for help in tumor collection. We acknowledge the financial support by the Luxembourg Institute of Health, the Fonds National de la Recherche of Luxembourg (AFR grant to AD: #5778172 – PhD2013-1/BM), the Fondation du Pélican de Mie et Pierre Hippert-Faber (Fondation de Luxembourg), Sächsisches Staatsministerium für Wissenschaft und Kunst (SMWK) project INTERDIS-2, Deutsche Krebshilfe and DFG-SFB-TRR79 project M8.

## Author contributions

Conceptualization: A.G., R.B., and S.P.N; methodology: A.D., A.G., N.H.C.B, S.P, A.D., T. B., and A.V.-B; Investigation: A.D., A.G., S.P., D.S, M.S., D.S., and A.M., Formal analysis: T.B., P.V.N. A.M., S. P., S.L., N.S., and S.F., Resources: C.H-M., F.H., R.B., and S.P.N., Supervision: A.G., F.A., A.S., A. D., A.V-B., and S.P.N. Writing—Original Draft: A.G. and S.P.N., Writing—Review & Editing: all authors.

## Additional information

**Competing interests:** The authors declare no competing interests.

