## [Peer Review File · Nature Communications]

Reviewers' comments:

Reviewer #1 (Remarks to the Author):

Dr Niclou and colleagues have attempted to address important and fundamental questions in regards to stem cell associated heterogeneity in glioblastoma and the impact of phenotypic plasticity.

As thoroughly discussed the issue of a predetermined hierarchical architecture related to the cancer stem cell has served as a rationale for targeting the cells with a variety of therapeutic endeavors. If in fact it is clearly demonstrated that the cellular subset is plastic and able to undergo change related to selection pressures such an approach is fundamentally flawed. As such the authors have attempted a series of experiments in glioblastoma to address this concern.

The authors move from examining in silico data in a TCGA, through flow cytometry based experiments to determine the heterogeneity of putative stem cell markers. I believe they convincingly demonstrate that there is both inter and intra-heterogeneity of these markers. Next through a mix of complex cell cultures under selection pressure and as best as possible by purifying subpopulations of cancer stem cells the authors show that glioblastoma cells in the subpopulations can freely undergo transitions and through creating stress such as hypoxia and exposure to micro-environmental cues that these cells are able to transition and adapt, repopulating toward a heterogeneous state.

In my view these experiments are convincing however limited by the usual in vivo and in vitro selection biases and limitations.

As the authors discuss and as demonstrated in the culture systems phenotypic plasticity is a powerful mechanism for overcoming selection pressures and as such if membrane marker defined CSC fractions are able to exploit such evolutionary changes in response to therapies these data do caution against eliminating tumors by targeting such cellular fractions.

The paper is well written, whilst dense and heavy going.

Reviewer #2 (Remarks to the Author):

This is a well written and well presented article focussed on the key concept that cancer stem cells, specifically in GBM, do not follow the classic hierarchical model but rather are an emergent property of the dialogue between tumor cell and context. This implicitly means that cancer stem cell like behavior can be achieved by any cancer cell given the appropriate context, if true this plastic cell state has important implications for cancer stem cell targeted treatments.

I thoroughly enjoyed reading the paper as it integrates a plethora of experimental approaches, including mathematical modeling, to present a strong case against the hierarchical model of GBM cancer stem cells. However, I felt it was lacking on key details and justifications specifically in regards to the choice and motivation of mathematical model and the choice contexts. I realize that there are space limitations and much of the methodological details are moved to the materials and methods or supplement. While I see this paper as being potentially suitable for Nature Communications, the following points (both major and minor) will need to be addressed before it can move forward.

Alexander R. A. Anderson.

Major Points

(i) Why use a Markov chain approach? As the authors explicitly state "cell state alterations occur due to stochastic cell state transitions only depending on the current state of the cell and possibly the experimental environment (e.g. hypoxia) and that proliferation rates of the involved phenotypes are approximately equal". Where is the justification for all of these assumptions?

(ii) What is the motivation for assuming that cell-cell interactions can be completely ignored?

(iii) These approaches do not consider the fact that cells will modify their context and as a result modify their potential states in a temporal manner. Why is this feedback not considered important especially when the creation and emergence of the niche seems key?

(iv) Space is also completely ignored even though hypoxia is certainly a spatial phenomena.

(v) In regards to context there has already been work that has shown the key role of pH in regulating stem like properties and it is know that cancer cells (GBM in particular) are highly glycolytic - a direct byproduct of glycolytic metabolism is lactic acid. No discussion is given as to the pH of the cultures, PDOXs, or patient samples and yet this could be a key driver of the observed phenomena that hypoxia would only further serve to exacerbate.

(vi) The resistance results are interesting as it implies there is no selection as a result of treatment which is very surprising! However is this tied to the fact that only low dose strategies are being used? Even if its not a stem related mechanism, I would have expected the heterogeneity to change under treatment with specific states being selected for?

Minor Points

(i) Whilst figure 2D is a work of art, its difficult to interpret the temporal dynamics in such a compressed figure. It would be useful to see this data displayed in a different manner.

(ii) Could the authors please clarify why lineage tracing approaches could not be used.

(iii) Others have tackled this plasticity perspective that have not been cited in this manuscript, please include the following papers:

1. H. Enderling, "Cancer stem cells: Small subpopulation or evolving fraction?," *Integr. Biol.*, vol. 7, no. 1, pp. 14–23.

2. J.G.Scott et al., "Microenvironmental variables must influence intrinsic phenotypic parameters of cancer stem cells to affect tumourigenicity," *PLoS Comput. Biol.*, vol. 10, no. 1, pp. 1–7.

3. N. Picco, et al, (2016) Niche-Driven Stem Cell Plasticity and its Role in Cancer Progression. *IEEE Trans. Biomed. Eng.*, March 2017; 64 (3): 528–537.

4. J.G. Scott et al (2017) Recasting the cancer stem cell hypothesis: unification using a continuum model of microenvironmental forces. <https://doi.org/10.1101/169615>.

RESPONSE TO REVIEWERS

Reviewer #1 (Remarks to the Author):

Dr Niclou and colleagues have attempted to address important and fundamental questions in regards to stem cell associated heterogeneity in glioblastoma and the impact of phenotypic plasticity.

As thoroughly discussed the issue of a predetermined hierarchical architecture related to the cancer stem cell has served as a rationale for targeting the cells with a variety of therapeutic endeavors. If in fact it is clearly demonstrated that the cellular subset is plastic and able to undergo change related to selection pressures such an approach is fundamentally flawed. As such the authors have attempted a series of experiments in glioblastoma to address this concern.

The authors move from examining in silico data in a TCGA, through flow cytometry based experiments to determine the heterogeneity of putative stem cell markers. I believe they convincingly demonstrate that there is both inter and intra-heterogeneity of these markers. Next through a mix of complex cell cultures under selection pressure and as best as possible by purifying subpopulations of cancer stem cells the authors show that glioblastoma cells in the subpopulations can freely undergo transitions and through creating stress such as hypoxia and exposure to micro-environmental cues that these cells are able to transition and adapt, repopulating toward a heterogeneous state.

In my view these experiments are convincing however limited by the usual in vivo and in vitro selection biases and limitations.

As the authors discuss and as demonstrated in the culture systems phenotypic plasticity is a powerful mechanism for overcoming selection pressures and as such if membrane marker defined CSC fractions are able to exploit such evolutionary changes in response to therapies these data do caution against eliminating tumors by targeting such cellular fractions.

The paper is well written, whilst dense and heavy going.

We thank Reviewer 1 for the positive evaluation and appreciation of our work. As highlighted by the reviewer our data caution against therapeutic strategies that specifically aim to target a limited subpopulation of putative cancer stem cells, which we believe is relevant for future clinical applications.

Reviewer #2 (Remarks to the Author):

This is a well written and well presented article focussed on the key concept that cancer stem cells, specifically in GBM, do not follow the classic hierarchical model but rather are an emergent property of the dialogue between tumor cell and context. This implicitly means that cancer stem cell like behavior can be achieved by any cancer cell given the appropriate context, if true this plastic cell state has important implications for cancer stem cell targeted treatments.

I thoroughly enjoyed reading the paper as it integrates a plethora of experimental approaches, including mathematical modeling, to present a strong case against the hierarchical model of GBM cancer stem cells. However, I felt it was lacking on key details and justifications specifically in regards to the choice and motivation of mathematical model and the choice contexts. I realize that there are space limitations and much of the methodological details are moved to the materials and methods

or supplement. While I see this paper as being potentially suitable for Nature Communications, the following points (both major and minor) will need to be addressed before it can move forward.

Alexander R. A. Anderson.

We are grateful to Reviewer 2 for the positive assessment of our manuscript and the recognition of the importance of mathematical modeling in elucidating state transitions and the non-hierarchical organization of the analyzed phenotypic states. Following the suggestion of the Reviewer we have now included more details and justifications of the modeling approach in the main manuscript text as detailed below.

Major Points

(i) Why use a Markov chain approach? As the authors explicitly state "cell state alterations occur due to stochastic cell state transitions only depending on the current state of the cell and possibly the experimental environment (e.g. hypoxia) and that proliferation rates of the involved phenotypes are approximately equal". Where is the justification for all of these assumptions?

We thank Reviewer 2 for highlighting these important aspects with regard to the modeling approach. The cell state compositions can be determined by two main processes: (1) stochastic state transitions due to cellular plasticity and (2) different proliferation rates reflecting selection. We experimentally demonstrated that all subpopulations proliferate at a similar rate both in normoxia (Figure 2C) and in hypoxia (Figure 3C). Hence, proliferation rates could be neglected for the estimation of transition rates. We agree with the Reviewer that beside the recent state of the cell, stochastic state transitions may potentially depend on the extracellular environment of the cell. Therefore, we choose to perform *in vitro* experiments in two different microenvironmental conditions, using normoxia and hypoxia as examples of relevant tumor microenvironments. For both of these conditions we estimated the transition rates separately, such that the dependence on extracellular conditions was taken into account in our model. These explanations have now been included in the manuscript (page 9).

Taking these considerations into account, we developed a stochastic compartment model represented by a Markov chain model. This model class is an established tool for analyzing cell state transitions from FACS-based experiments (see e.g. Gupta et al., *Cell* 2011, PMID: 21854987 and Buder et al. *Bioinformatics and biology insights*, 2017, PMID: 28659714). We acknowledge that the modeling may be simplistic in trying to extract as much insight as possible from as little assumptions as possible, however this is inherent to all available modeling approaches. Importantly however, we carefully verified that the model choice was applicable to our dataset and we were indeed able to validate several model predictions at the experimental level, e.g.:

- The steady state distribution predicted by the underlying Markov chain closely resembled the original tumor composition (see Figure 2F), although the data on the original tumor composition was not taken into consideration for the prediction. The model was purely based on the phenotypic composition of purified subpopulations at different time points. This observation has been included in the Results section (page 9)
- We have further designed cell compositions predicted to reach the equilibrium very fast *i.e.* Mix A and Mix B illustrated in Figure 2G. For these cell compositions, the model predicted 39 days to reach equilibrium, which could be validated experimentally, indicating that the Markov chain approach is able to reliably predict the time-scales of cell state transitions.
- Under hypoxic conditions, the Markov chain approach predicted that the phenotypic state "P10" represents a transient state which reflects the experimentally observed low proportions of P10, again validating the model prediction.

Taken together these data clearly validate the choice of the Markov chain approach. We added a more detailed justification of the modeling choice in the revised version of our manuscript in the Results (pages 9, 12) and Methods sections (page 35).

(ii) What is the motivation for assuming that cell-cell interactions can be completely ignored?

With regard to cell-cell interactions, we agree with the Reviewer that they most likely play a relevant role in state transitions. However in the absence of known quantifiable data about inter-cellular interactions, we neglect these components to keep the model tractable and focused on key processes.

(iii) These approaches do not consider the fact that cells will modify their context and as a result modify their potential states in a temporal manner. Why is this feedback not considered important especially when the creation and emergence of the niche seems key?

(iv) Space is also completely ignored even though hypoxia is certainly a spatial phenomena.

We thank the Reviewer for highlighting the importance of the temporal and spatial dimension in cellular adaptation processes. We agree that the creation and emergence of the niche *in vivo* crucially depends on the cellular context and that a corresponding feedback potentially plays an important role in the temporal development of the tumor. Similarly many inter-cellular mechanisms depend on the spatial arrangement of cells, especially oxygen supply and space is an important parameter for modeling the *in vivo* tumor environment. However, modeling of all existing *in vivo* interactions is currently not feasible, since not all parameters are known and/or can be quantified. Therefore our modeling is focused on *in vitro* experiments, where a limited number of parameters can be considered and experimentally controlled, e.g. to model the response to hypoxia, all GBM cells were subjected to low oxygen level (<0.5% O₂). This allowed us to examine in detail the exact response of cells to severe hypoxia, corresponding to *in vivo* niches around the necrotic tumor core (i.e. pseudopalisading hypoxic cells). This severe hypoxic niche has been highlighted in the literature as exerting a relevant selective pressure on putative cancer stem cells. This is now clarified in the Result section (page 10). Moreover, spatial aspects could be neglected in our experiments since cellular heterogeneity was present at the single sphere level without an obvious localization pattern (Fig. 1G). We added a discussion of these points to the revised version in the Results and Material and Methods sections (pages 9, 12, 35).

(v) In regards to context there has already been work that has shown the key role of pH in regulating stem like properties and it is know that cancer cells (GBM in particular) are highly glycolytic - a direct byproduct of glycolytic metabolism is lactic acid. No discussion is given as to the pH of the cultures, PDOXs, or patient samples and yet this could be a key driver of the observed phenomena that hypoxia would only further serve to exacerbate.

We thank the Reviewer for this important remark. We are aware of the key role of pH in affecting stem cell properties, therefore the pH of our cultures was regularly monitored by using media containing phenol red. Particular attention was given to the hypoxic cultures, known to decrease pH levels faster due to the glycolytic nature of the cells i.e. the release of lactic acid, as noted by the Reviewer. To avoid acidification of the hypoxic cultures, the medium was regularly replaced with fresh media pre-equilibrated in hypoxia. Although this allowed to minimize the influence of low pH during our experiments, we recognize that acidification is an inherent feature of severe hypoxia that cannot be excluded. We have included the information of pH monitoring in the revised manuscript (Results and Methods sections, pages 10, 30-31) and further discussed the link between pH, glycolysis and hypoxia (Discussion, page 18).

With regard to the *in vivo* situation, GBM cells in patient tumors or PDOX models are indeed subjected to different microenvironments, including gradients of hypoxia and pH levels. Our PDOX models reflected different microenvironmental landscapes of GBM, with some displaying a highly infiltrative phenotype with normal vasculature, while others showed signs of necrosis, hypoxia, vascular abnormalities and leakage. These characteristics have been extensively described in our previous publications (Keunen *et al.* *PNAS* 2011; Golebiewska *et al.* *BRAIN* 2014; Bougnaud *et al.* *Oncotarget* 2016). However with regard to CSC marker expression, we did not observe a clear correlation of markers with the histopathological phenotype (e.g. necrotic versus invasive landscape), suggesting that the presence of hypoxia is not the only driver of stem cell properties. These observations have now been added to the revised manuscript (page 6).

(vi) The resistance results are interesting as it implies there is no selection as a result of treatment which is very surprising! However is this tied to the fact that only low dose strategies are being used? Even if its not a stem related mechanism, I would have expected the heterogeneity to change under treatment with specific states being selected for?

We have applied a low concentration of TMZ in order to be under conditions that best resemble clinically relevant doses. However we also did not detect changes at the gene expression level of assessed markers in cells subjected to high TMZ doses (Table S6). We agree with the Reviewer that high TMZ doses applied over a prolonged time period to cultured cells may lead to a stronger selective pressure and the selection of certain phenotypic states. This condition, however, is unlikely to be achieved in patients with the current treatment protocol applied in the clinics, where the therapeutically achieved dose of TMZ in the brain is estimated at 5-50 μ M (see *e.g.* Ostermann *et al.*, 2004 PMID:15173079). We have adapted the text in the Result (page 16) and Discussion sections to better highlight this point (page 19).

Minor Points

(i) Whilst figure 2D is a work of art, its difficult to interpret the temporal dynamics in such a compressed figure. It would be useful to see this data displayed in a different manner.

We agree with the Reviewer that displaying state transitions for 16 subpopulations in time is not an easy task and may be difficult to interpret. The revised manuscript contains a more detailed explanation of the data displayed in Figure 2D (page 8-9) and we have also included an alternative more standard representation in Fig. S5A.

(ii) Could the authors please clarify why lineage tracing approaches could not be used.

Our manuscript is solely focused on phenotypic heterogeneity based on cell membrane markers associated with GBM CSCs. Lineage tracing techniques such as those based on genetic barcodes, are generally used to assess cellular hierarchy with regard to survival and proliferative potential, i.e. a functional cellular property (e.g. Lan *et al.*, 2017, PMID: 28854171). Our work is therefore not contradicting in any way the lineage tracing experiments, but reveals another aspect of GBM behavior. Unfortunately, current barcoding technologies do not allow for combined interrogation of genetic barcodes and multicolor phenotyping, because 1. barcoding based on DNA sequencing requires DNA isolation of bulk samples (thus destroying the cells); and 2. while fluorescence based-barcoding interferes with multicolor phenotyping on FACS as it requires at least 3 fluorescent barcodes with wide emission spectra, thus interfering with fluorochrome-conjugated antibodies against cell membrane markers and live/dead staining.

(iii) Others have tackled this plasticity perspective that have not been cited in this manuscript, please include the following papers:

1. H. Enderling, "Cancer stem cells: Small subpopulation or evolving fraction?," *Integr. Biol.*, vol. 7, no. 1, pp. 14–23.
2. J.G.Scott et al., "Microenvironmental variables must influence intrinsic phenotypic parameters of cancer stem cells to affect tumorigenicity," *PLoS Comput. Biol.*, vol. 10, no. 1, pp. 1–7.
3. N. Picco, et al, (2016) Niche-Driven Stem Cell Plasticity and its Role in Cancer Progression. *IEEE Trans. Biomed. Eng.*, March 2017; 64 (3): 528–537.
4. J.G. Scott et al (2017) Recasting the cancer stem cell hypothesis: unification using a continuum model of microenvironmental forces. <https://doi.org/10.1101/169615>.

We thank the Reviewer for highlighting these important papers. We have included the available publications in the final version of the manuscript (page 5). It should be noted though that we have passed the limit of references allowed by the journal.

REVIEWERS' COMMENTS:

Reviewer #2 (Remarks to the Author):

The authors have done an excellent job of addressing most of my concerns and clarified some of the others. I believe with the following relatively minor clarifications this manuscript is suitable for publication in Nature Communications.

It is important that the reader understands the underlying assumptions of the mathematical model and would appreciate the inclusion of statement regarding why cell interactions are not included even though they might be important in the main text.

Regarding the spatial aspects, the justification for neglecting it, is given by the observed heterogeneity at the single sphere level, does make some sense. However, from figure 1G its clear that all the proliferative action is on the boundary of the sphere (Ki67 staining) and the other markers seem to be distributed throughout. The proliferation behavior sets up an edge vs core variation that will change with tumor size. Does the size of the tumor significantly alter the degree of heterogeneity observed in the other markers?

In terms of context I agree that controlled experiments help us to better understand specific perturbations but even in these controlled conditions gradients exist. Regardless, I appreciate the authors explicitly discussing the potential impacts of context in the revised manuscript but would like further detail. Please add a sentence on the dynamic feedback of cells and their in vivo niche and in order to understand this will require the development of mathematical models that explicitly include such interactions.

Minor

- (i) Add a full stop midway through line 5 on page 4.
- (ii) Add a full stop after the bracket, line 5, page 9.
- (iii) 4 lines from the bottom of page 10, replace concordantly with concordant.

Alexander R. A. Anderson

REVIEWERS' COMMENTS:

Reviewer #2 (Remarks to the Author):

The authors have done an excellent job of addressing most of my concerns and clarified some of the others. I believe with the following relatively minor clarifications this manuscript is suitable for publication in Nature Communications.

We would like to thank the Reviewer for the positive evaluation of our manuscript and the valuable suggestions throughout the review process.

It is important that the reader understands the underlying assumptions of the mathematical model and would appreciate the inclusion of statement regarding why cell interactions are not included even though they might be important in the main text.

We have now included in the main text the following statement (page 10):

We acknowledge that cell-cell interactions are likely to play a role in state transitions. However, since marker heterogeneity was present at the single sphere level without a specific distribution pattern (Figure 1G) and in the absence of known quantifiable data about inter-cellular interactions, we neglected spatial aspects to keep the model tractable and focused on key processes.

Regarding the spatial aspects, the justification for neglecting it, is given by the observed heterogeneity at the single sphere level, does make some sense. However, from figure 1G its clear that all the proliferative action is on the boundary of the sphere (Ki67 staining) and the other markers seem to be distributed throughout. The proliferation behavior sets up an edge vs core variation that will change with tumor size. Does the size of the tumor significantly alter the degree of heterogeneity observed in the other markers?

We thank the Reviewer for pointing out this important point. The average sphere size is around 300 μm , although it may vary from very small clusters of about 10 cells up to large spheres of $>500\mu\text{m}$. In our analysis we included spheres of different sizes and did not observe an influence of sphere size on marker distribution. This statement is now added to the Methods section (page 23).

In terms of context I agree that controlled experiments help us to better understand specific perturbations but even in these controlled conditions gradients exist. Regardless, I appreciate the authors explicitly discussing the potential impacts of context in the revised manuscript but would like further detail. Please add a sentence on the dynamic feedback of cells and their in vivo niche and in order to understand this will require the development of mathematical models that explicitly include such interactions.

We agree that further development of mathematical modeling will be crucial to understand the dynamic interactions within the tumor microenvironment. We have included this statement in the Discussion (page 21).

Minor

- (i) Add a full stop midway through line 5 on page 4.
- (ii) Add a full stop after the bracket, line 5, page 9.
- (iii) 4 lines from the bottom of page 10, replace concordantly with concordant.

Corrected

Alexander R. A. Anderson